# PARP14 and PARP9/DTX3L regulate interferon-induced ADP-ribosylation

Pulak Kar[1,2,8], Chatrin Chatrin[1,8], Nina Đukić [1,8], Osamu Suyari [1], Marion Schuller[1], Kang Zhu [1], Evgeniia Prokhorova[1], Nicolas Bigot [3], Domagoj Baretić[1], Juraj Ahel [4], Jonas Damgaard Elsborg [5], Michael L Nielsen[5], Tim Clausen[4,6], Sébastien Huet[3], Mario Niepel [7], Sumana Sanyal [1], Dragana Ahel[1], Rebecca Smith [1✉] & Ivan Ahel [1✉]

## Abstract

**PARP-catalysed ADP-ribosylation (ADPr) is important in regulating various cellular pathways. Until recently, PARP-dependent mono-ADP-ribosylation has been poorly understood due to the lack of sensitive detection methods. Here, we utilised an improved antibody to detect mono-ADP-ribosylation. We visualised endogenous interferon (IFN)-induced ADP-ribosylation and show that PARP14 is a major enzyme responsible for this modification. Fittingly, this signalling is reversed by the macrodomain from SARS-CoV-2 (Mac1), providing a possible mechanism by which Mac1 counteracts the activity of antiviral PARPs. Our data also elucidate a major role of PARP9 and its binding partner, the E3 ubiquitin ligase DTX3L, in regulating PARP14 activity through protein-protein interactions and by the hydrolytic activity of PARP9 macrodomain 1. Finally, we also present the first visualisation of ADPr-dependent ubiquitylation in the IFN response. These approaches should further advance our understanding of IFN-induced ADPr and ubiquitin signalling processes and could shed light on how different pathogens avoid such defence pathways.**

**Keywords** Immune Response; Interferon Response; ADP-ribosylation; Ubiquitin; SARS-CoV2
**Subject Categories** Immunology; Post-translational Modifications & Proteolysis

See also: VC Ribeiro et al

## Introduction

Cells utilise post-translational modifications (PTM) to regulate dynamic signalling events. The rapid and transient nature of some modifications allow efficient transmission of signals in response to internal or external stimuli, including viral infections. These modifications alter the physical properties of their target molecules, which in turn determine their fate and function. ADP-ribosylation (ADPr) is a reversible modification found in organisms from all kingdoms of life including viruses and can be covalently attached to diverse amino acid residues on proteins as well as on nucleic acids (Fontana et al, 2023; Groslambert et al, 2021; Gupte et al, 2017; Perina et al, 2014). Each PTM has dedicated machineries to encode and decode the modification. In humans, ADP-ribosylation is primarily catalysed by a family of 17 enzymes known as PARPs: PARP1-4, PARP6-16, and TNKS1/2 which are also known as PARP5a/5b (Lüscher et al, 2022). All PARPs share an ADP-ribosyl transferase (ART) catalytic domain, which hydrolyses NAD$^+$ and transfers ADP-ribose onto a target, releasing nicotinamide in the process. Four PARPs (PARP1, 2, TNKS1, TNKS2) catalyse long poly-ADP-ribose (PAR) chain formation while another 11 members (PARP3, 4, 6, 7, 8, 10, 11, 12, 14, 15, 16) catalyse mono-ADP-ribosylation (MAR). Two PARPs, PARP9 and PARP13, are catalytically inactive (Suskiewicz et al, 2023; Vyas et al, 2014). ADP-ribosylated molecules can be recognised by a number of ADPr-binding domains (readers) and also enzymes that reverse ADPr modification (erasers). PAR and MAR signals are recognised by several protein modules including PAR-binding motifs (PBM), PAR-binding Zinc-finger (PBZ), WWE domains that recognises iso-ADPr subunit, and the macrodomain fold that recognises the terminal ADP-ribose moiety (Ahel et al, 2008; DaRosa et al, 2015; Karras et al, 2005; Teloni and Altmeyer, 2016). Finally, ADPr signal removal is achieved by ADP-ribosylhydrolases (ARHs), hydrolytic macrodomains and NADAR domains (Rack et al, 2020a; Schuller et al, 2023a).

Several human PARPs have been implicated in immunity and antiviral response, with their expression induced by interferons. These interferon-induced PARPs include PARP7, 9, 10, 11, 12, 13, 14, 15 (Grunewald et al, 2019). Collectively, these PARPs have been shown to provide resistance against various RNA viruses, such as HIV, influenza, Ebola, and coronaviruses, with the exception of

[1]Sir William Dunn School of Pathology, University of Oxford, Oxford OX1 3RE, UK. [2]Department of Biological Sciences, SRM University-AP, Amaravati 522502, India. [3]Univ Rennes, CNRS, IGDR (Institut de génétique et développement de Rennes) - UMR 6290, BIOSIT – UMS3480, F-35000 Rennes, France. [4]Research Institute of Molecular Pathology (IMP), Vienna BioCenter, Vienna, Austria. [5]Proteomics Program, Novo Nordisk Foundation Center for Protein Research, Faculty of Health and Medical Sciences, University of Copenhagen, Blegdamsvej 3B, 2200 Copenhagen, Denmark. [6]Medical University of Vienna, Vienna, Austria. [7]Ribon Therapeutics, Cambridge, MA 02140, USA. [8]These authors contributed equally: Pulak Kar, Chatrin Chatrin, Nina Đukić. ✉E-mail: rebecca.smith@path.ox.ac.uk; ivan.ahel@path.ox.ac.uk

PARP7 which has been reported to also have proviral activities (Grunewald et al, 2019; Kerns et al, 2008; Müller et al, 2007; Voth et al, 2021; Yamada et al, 2016; Zhu et al, 2011). Among these antiviral PARPs, PARP14 has been suggested to be the main PARP protecting against coronavirus infection using a murine coronavirus model (Grunewald et al, 2019). PARP14 also has important roles in mediating inflammation (Schenkel et al, 2021) as well as in the maintenance of genome stability (Dhoonmoon and Nicolae, 2023; Nicolae et al, 2015).

PARP14 ADP-ribosylates a number of protein targets, with in vitro data also showing the ability to modify nucleic acids (Carter-O'Connell et al, 2018; Đukić et al, 2023; Suskiewicz et al, 2023), however, the sites and physiological consequences of these modifications are not known. Intriguingly, PARP14 also has a hydrolytic macrodomain 1 that was shown to reverse PARP14-mediated ADP-ribosylation (Delgado-Rodriguez et al, 2023; Đukić et al, 2023; Rack et al, 2020b; Torretta et al, 2023). This hydrolytic activity is also conserved in the closely related PARP9 macrodomain 1 (Delgado-Rodriguez et al, 2023; Đukić et al, 2023; Rack et al, 2020a; Torretta et al, 2023). Many viruses, including the members of *Coronaviridae, Togaviridae, Iridoviridae*, rubella virus, and hepatitis E virus, have evolved to encode a macrodomain to counteract the antiviral function of PARPs (Fehr et al, 2016; Fehr et al, 2018). Moreover, the Nsp3 macrodomain 1 (Mac1) of SARS-CoV-2 (SARS2) is most closely related to the first macrodomains of PARP14 and PARP9 (Đukić et al, 2023; Rack et al, 2020b). The recombinant coronaviral Mac1 and Mac1-like proteins from other viruses possess hydrolytic activity that can cleave the protein-ADP-ribose and nucleic acid-ADP-ribose bond (Abraham et al, 2020; Đukić et al, 2023; Fehr et al, 2016; Li et al, 2016; Munnur and Ahel, 2017; Munnur et al, 2019; Rack et al, 2020b). Indeed, the three macrodomain-containing PARPs, PARP9, PARP14, and PARP15, are under positive evolutionary selection, strongly suggesting their engagement in an evolutionary arms race with viral pathogens (Daugherty et al, 2014; Delgado-Rodriguez et al, 2023).

PARP14 is the largest of all human PARPs with 1801 amino acid residues. In addition to its hydrolytic macrodomain, it is comprised of three RNA-recognition motif (RRM) domains, eight K-homology (KH) domains, and two additional non-catalytic ADP-ribose-binding macrodomains, a WWE domain, and a catalytically active ART domain which catalyses mono-ADP-ribosylation (Suskiewicz et al, 2023).

PARP14 seems to be closely interconnected to PARP9, and the PARP9-binding partner E3 ubiquitin ligase DTX3L (Ashok et al, 2022; Takeyama et al, 2003). PARP9 and DTX3L form a tight complex (Takeyama et al, 2003) that has suggested roles in DNA repair, as well as in the response to bacterial or viral infection (Huang et al, 2023; Thirunavukkarasu et al, 2023; Yan et al, 2023; Zhang et al, 2015) and carcinogenesis (Bachmann et al, 2014). PARP9 (854 aa) consists of two KH domains, two macrodomains, and a seemingly inactive ART domain containing a QYT motif instead of an active HYX motif (Aguiar et al, 2005; Vyas et al, 2014). As with the first macrodomain of PARP14, the first macrodomain of PARP9 is an active ADPr hydrolase with activity towards Glu/Asp-ADPr and ADP-ribosylated ssDNA or ssRNA (Delgado-Rodriguez et al, 2023; Đukić et al, 2023; Torretta et al, 2023). However, the macrodomains of PARP9 were also proposed as Cys-ADPr readers (Yang et al, 2021) and a viral dsRNA sensor (Xing et al, 2021). Meanwhile, the PARP9-binding partner, DTX3L

(740 aa), consists of one RRM domain, five KH domains, a RING domain with a ubiquitin E3 ligase activity (Takeyama et al, 2003), and a Deltex C-terminal (DTC) domain which binds to $NAD^+$ or ADP-ribose (Ahmed et al, 2020; Chatrin et al, 2020; Zhu et al, 2022). In a glance, PARP14 can be viewed to have the combined domain architecture of both PARP9 and DTX3L, only without the RING and DTC domain.

PARP14, PARP9 and DTX3L, are located in the same chromosome locus 3q21.1 (Aguiar et al, 2005; Juszczynski et al, 2006) where the DTX3L and PARP9 genes are arranged in a head-to-head orientation, sharing a single bidirectional interferon-gamma responsive promoter (Juszczynski et al, 2006). This coordinated gene expression regulation results in the formation of a biological PARP9/DTX3L complex, which shuttles between the nucleus and cytoplasm (Takeyama et al, 2003). PARP14 expression is likely also coordinated with PARP9/DTX3L as the protein levels of each are elevated upon both type I and II IFN stimulation (Iwata et al, 2016; Moore et al, 2021). Moreover, PARP14 has also been reported to interact with PARP9/DTX3L following LPS treatment in RAW264.7 cells (Caprara et al, 2018).

In this study, we visualised endogenous PARP14-dependent ADPr in cells, showing that PARP14 is a major enzyme responsible for IFN-induced ADPr in human cells. We also demonstrate that SARS2 Mac1 can remove this ADPr in cells. We further show that PARP14 activity is cross-regulated by DTX3L and PARP9, through both protein-protein interactions between the KH8-WWE-ART domains of PARP14 and the KH2-KH3 domains of DTX3L as suggested by our structural models and functional data, and through the hydrolytic activity of PARP9 macrodomain 1. Lastly, we provide evidence for ubiquitylation that is strictly dependent on PARP14-dependent ADPr during IFN response.

# Results

## PARP14 is a major ART activated after interferon stimulation

ADP-ribosylation is becoming increasingly implicated in the cellular response to immune stimuli. In particular, the importance of PARP14 in regulating immune responses has been the focus of a number of recent studies. To understand the extent of PARP14's contribution to endogenous ADPr, we used a newly derived mono-ADPr antibody (Longarini et al, 2023), which we have recently shown can efficiently detect PARP14-derived mono-ADPr in overexpression systems (Đukić et al, 2023). When we examined the mono-ADPr pattern of endogenous ADPr in A549 cells, we detected several ADPr bands, even in the absence of interferon stimulation (Figs. 1A and EV1A). In particular, we observed prominent ADPr of a protein approximately 50 kDa in size which is dependent on PARP14 catalytic ADP-ribosyl transferase (ART) activity as treatment of cells with a specific PARP14 ART inhibitor (PARP14i; RBN012759) resulted in the loss of this ADPr band, in addition to many others. Upon IFNγ stimulation, the ADPr signal at 50 kDa was greatly strengthened, in addition to several other proteins also showing increased ADPr, including a protein of around 65 kDa (Figs. 1A and EV1A). We also observed increased ADPr of a protein approximately 200 kDa in size, which corresponds to the size of PARP14. To examine this, we

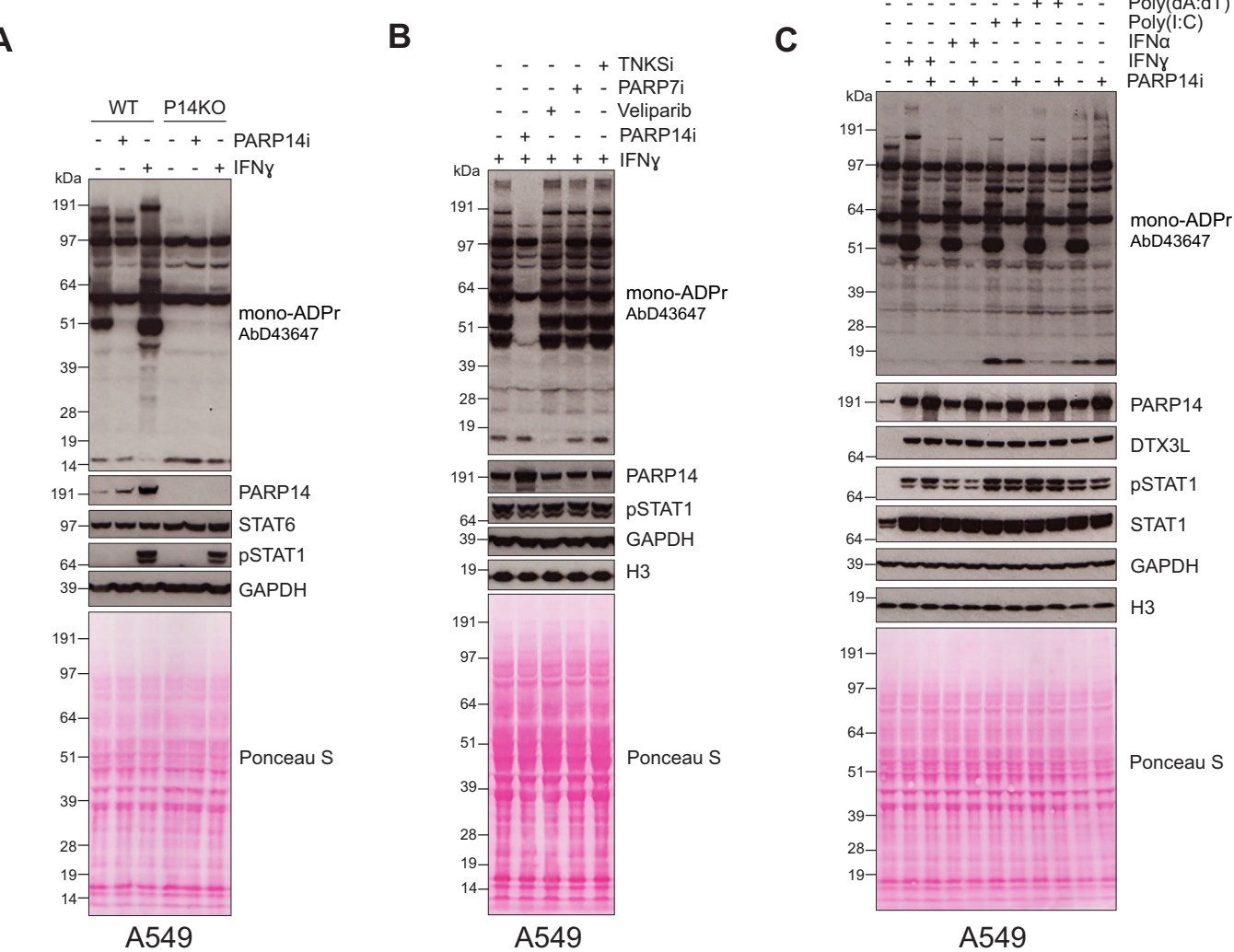

**Figure 1. Immunity responses induce PARP14-dependent ADP-ribosylation.**

(A) A549WT and PARP14 knockout cells untreated or treated with 0.5 μM PARP14i (RBN012759) or with IFNγ (100 ng/mL). Cell lysates were examined by western blot with the indicated antibodies. (B) A549 WT cells were treated with 0.5 μM PARP14i, 100 nM PARP7i, 1 μM Veliparib or 1 μM TNKSi and stimulated with IFNγ (100 ng/mL). Cell lysates were examined by western blot with the indicated antibodies. (C) A549 WT cells were treated with IFNγ (100 ng/mL), IFNα (1000 U/mL), poly(dA:dT), poly(I:C), or 5' triple phosphate dsRNA (PPP-dsRNA) in the presence or absence of 0.5 μM PARP14i. Cell lysates were examined by western blot with the indicated antibodies. For all western blots, pSTAT1 levels were used as a positive control for immune response activation. GAPDH was used as a loading control. Source data are available online for this figure.

immunoprecipitated PARP14 and examined its ADPr status (Fig. EV1B). Gratifyingly, we were able to detect mono-ADPr of PARP14 after IFNγ treatment that was lost upon addition of a PARP14 inhibitor, confirming that PARP14 is auto-modified upon IFNγ stimulation. A549 PARP14 knockout cells showed a similar mono-ADPr pattern to WT cells treated with PARP14 inhibitor and IFNγ stimulation did not result in an increase in ADPr (Fig. 1A). To further confirm these results, we treated IFNγ stimulated A549 WT cells with a PARP14 catalytic inhibitor (PARP14i, RBN012759) which resulted in a robust reduction of mono-ADPr signal, strongly suggesting that the majority of interferon induced ADPr in A549 cells is dependent on PARP14 catalytic activity (Fig. 1B).

Next, we aimed to investigate whether the observed increase in ADPr signal following interferon stimulation is dependent on some other PARP enzymes. To do so, we treated IFNγ stimulated cells with different PARP inhibitors including the PARP7 inhibitor (RBN-2397) and a Tankyrase inhibitor (XAV939). As expected, there was no major change in mono-ADPr signal, suggesting that these PARPs are not contributors to the robust ADPr levels seen in these conditions. Notably, treatment with the PARP1/2 specific inhibitor Veliparib resulted in the reduction in mono-ADPr of several proteins that were not modulated by IFN treatment. Finally, depletion of another antiviral PARP, PARP12 (Kerr et al, 2023), did not alter mono-ADPr after IFNγ-induction (Fig. EV1C,D). In order to confirm that increased ADPr signature is not limited to IFNγ treatment, we treated

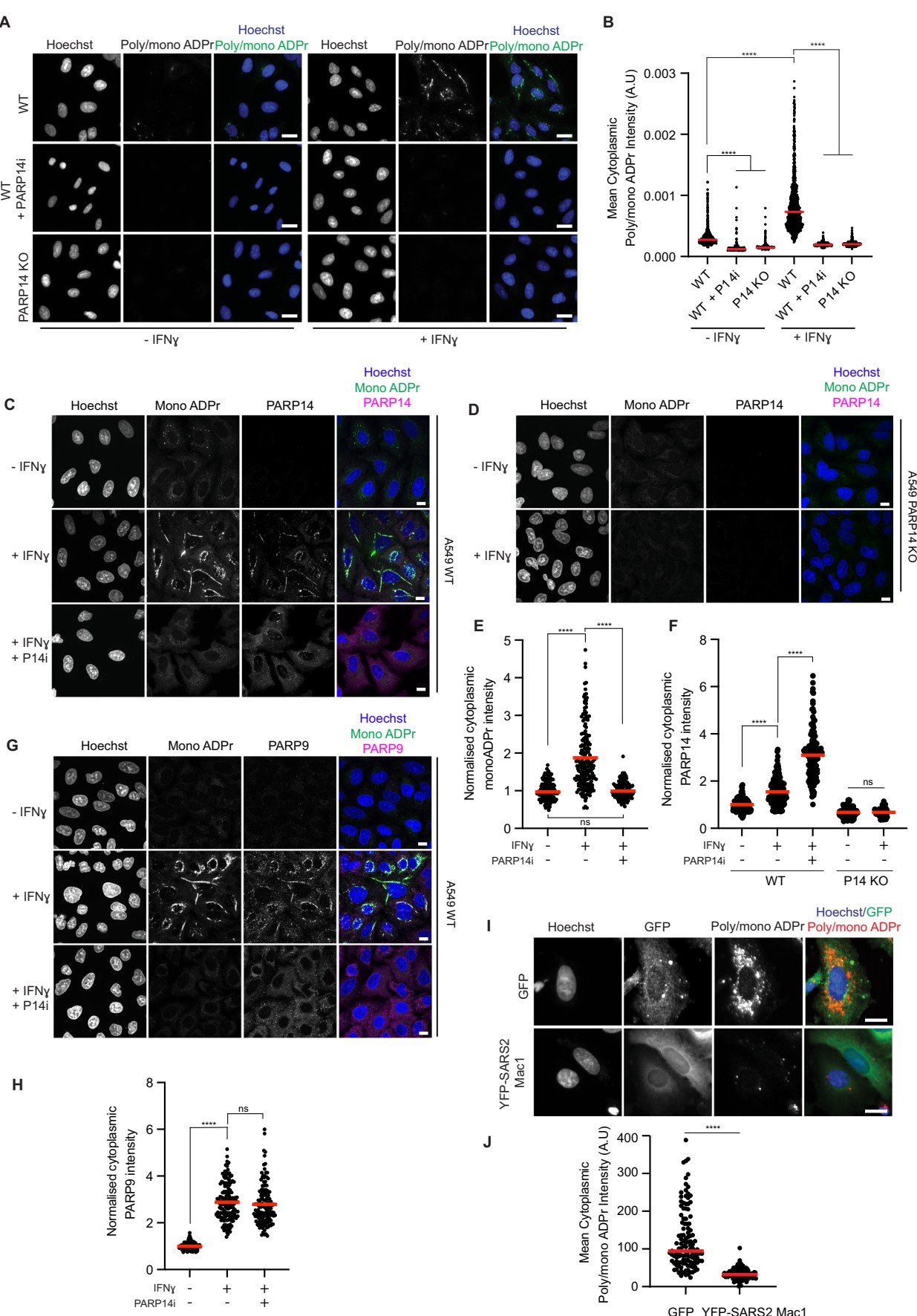

Figure 2. PARP14 increases cytoplasmic ADPr after IFNγ stimulation.

(A) Widefield images showing A549 WT or PARP14 KO cells untreated or treated with IFNγ in the presence or absence of PARP14i (0.5 μM). Cells were stained with Hoechst (Blue) and ADPr (poly/mono-ADPr antibody, CST #83732) (Green). Scale bar = 20 μm. (B) Mean intensity of cytoplasmic signal from (A) measured in arbitrary units (A.U). The red line indicates mean grey values. (C) Confocal images showing A549 WT cells untreated or treated with IFNγ (100 ng/mL) in the presence or absence of PARP14i (0.5 μM). Cells were stained with Hoechst (Blue), mono-ADPr (AbD43647 IgG-coupled antibody) (Green) and PARP14 (Magenta). Scale bar = 20 μm. (D) Confocal images showing A549 PARP14 KO cells untreated or treated with IFNγ (100 ng/mL). Cells were stained with Hoechst (Blue), mono-ADPr (AbD43647 IgG-coupled antibody) (Green) and PARP14 (Magenta). Scale bar = 20 μm. (E) Normalised intensity of cytoplasmic mono-ADPr signal from (C). Fluorescence intensity was normalised to WT cells in the absence of IFNγ or PARP14i. Red line indicates mean values. (F) Normalised intensity of PARP14 cytoplasmic signal from (C, D). Fluorescence intensity was normalised to WT cells in the absence of IFNγ or PARP14i. Red line indicates mean grey values. (G) Confocal images showing A549 WT cells untreated or treated with IFNγ (100 ng/mL) in the presence or absence of PARP14i (0.5 μM). Cells were stained with Hoechst (Blue), mono-ADPr (AbD43647 IgG-coupled antibody) (Green) and PARP9 (Magenta). Scale bar = 20 μm. (H) Normalised intensity of cytoplasmic PARP9 signal from (G). Fluorescence intensity was normalised to WT cells in the absence of IFNγ or PARP14i. Red line shows the mean. (I) Widefield images showing A549 WT cells expressing GFP or YFP-Nsp3 Mac1 treated with IFNγ (100 ng/mL). Cells were stained with Hoechst (Blue), GFP/YFP (Green) and ADPr (poly/mono-ADPr antibody, CST #83732) (Red). Scale bar = 20 μm. (J) Mean intensity of cytoplasmic signal from (I) measured in arbitrary units (A.U). Red line indicates mean grey values. For all graphs, statistical analysis was determined using a Kruskal Wallis test with Bonferroni correction followed by post-hoc Dunn's test. Asterisks indicate statistical significance (ns: not significant, ****$p < 0.0001$). Data information: Graphs in (B), (E), (F), (H) and (J) show individual cell measurements with a minimum of 108 cells per condition. The red line indicates the mean. For (E), (F) and (H), the intensity measurement was normalised to the mean intensity for WT cells without PARP14i or IFNγ treatment. Data are representative of a minimum of three independent replicates. Source data are available online for this figure.

A549 WT cells with other immune stimuli including IFNα and DNA or RNA mimics poly(dA:dT) and poly(I:C)/5' phosphate dsRNA respectively (Fig. 1C). As with IFNγ, we were able to see a strong increase in mono-ADPr with each stimuli that was inhibited by treatment with PARP14i. Together, these results show that PARP14 ART activity is activated in response to a wide range of immune stimuli, and that its ART activity is a major contributor in the production of mono-ADPr in these conditions.

## Interferon and PARP14 ADPr activity regulate PARP14 protein levels

Notably, we observed an increase in PARP14 protein levels when cells were treated with IFNγ and other stimuli (Fig. 1A–C), consistent with previous data showing PARP14 as one of the readily interferon-inducible genes and its role as an interferon-stimulated PARP (Caprara et al, 2018; Lüscher et al, 2022; Moore et al, 2021). Thus, one of the mechanisms for induction of PARP14 activity upon IFN treatment is upregulation of PARP14 protein levels. Moreover, we also observed the notable additional increase of PARP14 protein levels upon treatment with PARP14i (Fig. 1B,C), in line with previous reports (Schenkel et al, 2021), suggesting the presence of other levels of PARP14 regulation. The increase in PARP14 protein levels with PARP14i was also observed in unstimulated conditions (Fig. 1A) and suggests that PARP14 ADPr activity regulates its own stability. Fittingly, we observed that in an overexpression system, protein levels of a PARP14 mutant defective in its macrodomain 1 hydrolytic activity were lower compared to the WT counterpart despite identical expression conditions, and that this could be reversed by treatment with PARP14i or expression of SARS2 Nsp3 Mac1 which removes ADPr from PARP14 (Fig. EV1E,F) (Đukić et al, 2023). Altogether our data suggests that automodification of PARP14 controls PARP14 stability by an as yet unidentified mechanism.

## PARP14-dependent ADPr localises to structures within the cytoplasm

Next, we aimed to observe endogenous PARP14-dependent ADPr in a cellular context. For this, we treated WT or PARP14 knockout

(KO) A549 cells with IFNγ and examined changes to ADPr using immunofluorescence. We observed that there was a strong increase in cytoplasmic ADPr upon IFNγ stimulation using an antibody that recognises both mono- and poly-ADPr and that this was dependent on PARP14 activity (Fig. 2A,B). Inhibition of the catalytic activity with treatment of cells with PARP14i or loss of PARP14 in knockout cells, prevented the increase in cytoplasmic ADPr. Furthermore, when we examined ADPr with the mono-ADPr-specific antibody, we again saw an increase in cytoplasmic PARP14-dependent ADPr (Fig. 2C–E), suggesting that most of the IFN-induced ADPr is in the mono form. We also observed increased protein levels of both PARP14 and PARP9 (Figs. 2C–H and EV2A–C), and that both proteins colocalised with the cytoplasmic ADPr (Fig. 2C,G). Importantly, we found that expression of the hydrolytic macrodomain from SARS2 Nsp3 (Mac1) could reduce interferon-induced ADPr in A549 cells (Fig. 2I,J), in agreement with previous reports (Russo et al, 2021). Given that both PARP14 and PARP9 colocalise with the interferon-induced ADPr, we next investigated if there was any crosstalk between PARP9/DTX3L and PARP14.

## PARP14 and DTX3L work together to regulate ADPr

As previously mentioned, PARP14 is expressed from the same locus as PARP9/DTX3L (chromosome 3q21.1), and their expression and activity has been suggested to be cross-regulated (Bachmann et al, 2014; Caprara et al, 2018; Iwata et al, 2016). Additionally, an AlphaFold-predicted structure suggests a trimeric PARP9-DTX3L-PARP14 complex, where DTX3L tandem KH2-5 domains function as a central hub organising the complex. Specifically, the model shows DTX3L KH2-3 interacting with PARP14 KH8-WWE-ART and DTX3L KH4-5 interacting with PARP9 KH1-2-ART, while there is no direct contact between PARP9 and PARP14 (Figs. 3A,B and EV3A). Due to these predicted interactions, we wanted to examine if PARP9 and/or DTX3L would play a role in regulating PARP14-dependent ADPr in response to immune stimuli. Firstly, we depleted DTX3L in A549 cells and observed a difference in the PARP14-dependent ADPr signal (Fig. 3C). In particular, we observed an increase in mono-ADPr of a protein of ~65 kDa and a decrease in mono-ADPr of the ~50 kDa protein. Again, the ADPr

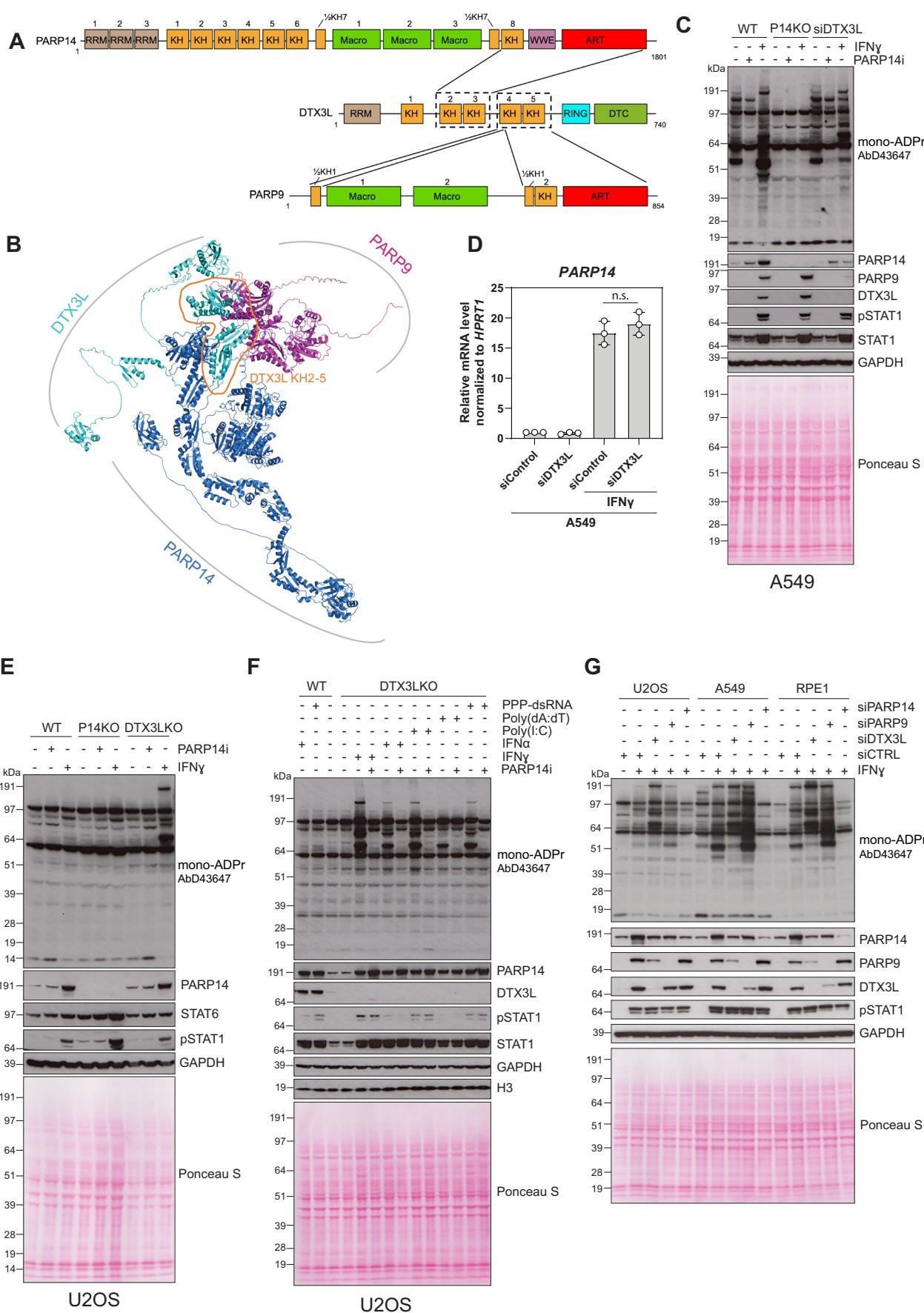

**Figure 3. PARP9/DTX3L regulate PARP14-dependent ADPr.**

(A) Schematic domain representation of PARP14, PARP9 and DTX3L. (B) AlphaFold prediction of PARP14 (blue), DTX3L (turquoise) and PARP9 (purple) interaction. DTX3L KH domains 2–5 are highlighted in orange. (C) A549 WT, PARP14 KO and DTX3L-depleted cells untreated or treated with 0.5 μM PARP14i or with IFNγ (100 ng/mL). Cell lysates were examined by western blot with the indicated antibodies. (D) The relative gene expression analysis of *PARP14* in unstimulated and IFNγ (100 ng/mL) stimulated A549 cells as determined by RT-qPCR, normalized to the expression of *HPRT1*. Error bars indicate average S.D. from three independent replicates. Asterisks indicate statistical significance compared with the control, as determined by Welch's t-test (ns: not significant, siCTRL vs siDTX3L +IFNγ $p = 0.3852$). (E) U2OS WT, PARP14 KO and DTX3L KO cells were treated with 0.5 μM PARP14i or with IFNγ (100 ng/mL). Cell lysates were examined by western blot with the indicated antibodies. (F) U2OS WT or DTX3L KO cells were treated with IFNγ (100 ng/mL), IFNα (1000 U/mL) poly(dA:dT), poly(I:C), or 5′ triple phosphate dsRNA (PPP-dsRNA) in the presence or absence of 0.5 μM PARP14i. Cell lysates were examined by western blot with the indicated antibodies. (G) U2OS, A549 and RPE1 cells depleted or not of DTX3L, PARP9 or PARP14 were treated with IFNγ (100 ng/mL). Cell lysates were examined by western blot with the indicated antibodies. For all western blots, pSTAT1 levels were used as a positive control for immune response activation. GAPDH was used as a loading control. Source data are available online for this figure.

signals were dependent on PARP14 ART activity as treatment with PARP14i resulted in a dramatic reduction in ADPr and was absent in PARP14 KO cells. We also observed that PARP14 protein levels did not reach the same levels upon depletion of DTX3L as in control A549 cells after interferon stimulation or PARP14i treatment, although this was not due to changes at a transcriptional level (Fig. 3D). Noteworthy, we observed that DTX3L depletion also affected the expression or stability of PARP9 as previously described (Bachmann et al, 2014). Interestingly, the apparent induction in ADPr signal with IFNγ treatment could not be observed in U2OS WT cells (Fig. 3E); however, the loss of DTX3L in U2OS cells resulted in an increase in ADPr signal, in particular at bands ~65 kDa and 200 kDa, the sizes of the major signals also observed in A549 cells (Figs. 3E and EV3B). As with A549 cells, we examined if a similar response is obtained with other immune stimuli, such as IFNα and DNA or RNA mimics. Here we also observed a PARP14-specific induction of ADPr, but only in DTX3L knockout cells (Fig. 3F). A similar phenomenon was observed in RPE1 cells, where an obvious increase in IFNγ- or poly(I:C)-induced ADPr signal was seen upon depletion of DTX3L (Figs. 3G and EV3C). These data suggest that DTX3L modulates PARP14 protein ADPr activity, possibly by direct binding as suggested in our predicted model. As DTX3L and PARP9 have been proposed to work together in regulating ubiquitination and ADPr (Yan et al, 2023; Yang et al, 2017; Zhang et al, 2015), we further investigated how depletion of PARP9 affected interferon-induced ADPr in A549, U2OS and RPE1 cells. In both A549 and RPE1 cells, depletion of PARP9 appeared to increase ADPr after IFN-induction or poly(I:C) treatment, with the major targets appearing to be the same 65 and 50 kDa bands, while depletion of PARP9 in U2OS cells had a modest effect on ADPr (Figs. 3G and EV3C). The increase in ADPr after PARP9 depletion suggests that PARP9 could be contributing to modulating ADPr levels with its recently identified hydrolytic macrodomain 1 (MD1) (Delgado-Rodriguez et al, 2023; Đukić et al, 2023; Torretta et al, 2023). Indeed, we found that overexpression of PARP9 WT, in contrast to PARP9 MD1 mutant with catalytically inactive macrodomain, dampened PARP14-mediated ADPr in 293T cells (Fig. EV3D).

## PARP14, PARP9 and DTX3L regulate the formation of ADPr and ubiquitin foci

Given that PARP14, PARP9 and DTX3L appear to differentially regulate cellular levels of ADPr, we decided to investigate how the cellular localisation of ADPr was altered under similar conditions. Firstly, we could again see strong cytoplasmic ADPr signals in A549

cells using the mono-ADPr specific antibody (Fig. 4A). Interestingly, when we depleted cells of PARP9, we were still able to observe strong cytoplasmic ADPr, however the foci had become smaller with a notably different distribution throughout the cytoplasm. Depletion of DTX3L also altered the cytoplasmic ADPr; however, in this case with a prominent decrease of the ADPr signal compared to both control and PARP9-depleted cells, but with some small ADPr foci still visible throughout the cytoplasm. Depletion of PARP14 resulted in a complete loss of cytoplasmic ADPr, as seen above (Fig. 2A), confirming that PARP14 is a major contributor to the IFN-induced mono-ADPr. Next, given that DTX3L is an active E3 ubiquitin ligase (Takeyama et al, 2003; Zhu et al, 2022) and that PARP9 and DTX3L have been reported to regulate ubiquitination in response to viral infection (Zhang et al, 2015), we also investigated if we could observe any changes to ubiquitination after IFN treatment. Strikingly, we observed colocalization of ADPr and ubiquitin in the cytoplasmic foci, and that this was dramatically increased upon depletion of PARP9 (Figs. 4B,C and EV4A,B). Furthermore, we observed the opposite effect upon depletion of DTX3L or PARP14. While some small ADPr foci remained, particularly for DTX3L depletion, the number of foci staining for both ADPr and ubiquitin was significantly diminished. Altogether, this data suggests a cross regulation between the three proteins where PARP14 can catalyse ADPr of different targets upon immune stimulation, DTX3L ubiquitinates PARP14-dependent ADPr or PARP14 ADP-ribosylated macromolecules and that this process is dampened by PARP9.

## PARP9 and DTX3L modulate PARP14 activity in vitro

Our results presented here, along with published data, suggest there are at least two possible mechanisms of how PARP9/DTX3L could modulate PARP14-dependent ADPr at the protein level: (1) removal of ADPr through hydrolysis or (2) inhibition of PARP14 activity through a direct interaction.

The first possible mechanism by which PARP9/DTX3L could modulate PARP14 is by removing PARP14 ADPr through the hydrolytic activity of PARP9 macrodomain 1, as suggested by previous studies (Delgado-Rodriguez et al, 2023; Đukić et al, 2023; Torretta et al, 2023). While testing this, we opted to use a minimal PARP14 WWE-ART fragment which lacks the hydrolytic macrodomain of PARP14, ensuring any hydrolytic activity would be a result of PARP9 MD1. To reconstitute this reaction, we first performed PARP14 WWE-ART auto-ADP-ribosylation with $^{32}$P-NAD$^+$ and used this as a substrate for subsequent hydrolysis (Rack et al, 2020b). Indeed, the hydrolytic macrodomains PARP9 MD1, PARP14 MD1, and SARS2 Mac1 all removed ADPr from

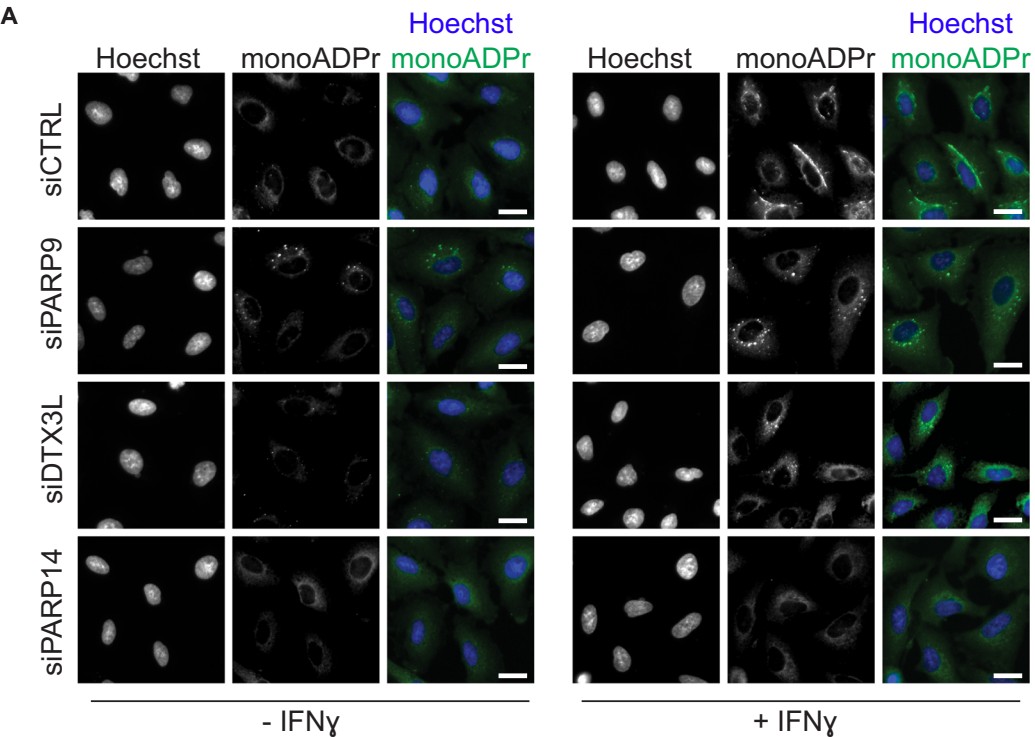

**A**

Hoechst | monoADPr | Hoechst monoADPr | Hoechst | monoADPr | Hoechst monoADPr

siCTRL
siPARP9
siDTX3L
siPARP14

− IFNγ                    + IFNγ

**B**

Hoechst | Poly/Mono ADPr | Ubiquitin | Hoechst Poly/Mono ADPr Ubiquitin | Zoom

siCTRL
siPARP9
siDTX3L
siPARP14

+ IFNγ

**C**

Ubiquitin / ADPr foci colocalisation per cell

siCTRL  siPARP9  siDTX3L  siPARP14

**Figure 4.   PARP14, PARP9 and DTX3L regulate ADPr and ubiquitin foci formation.**

(A) Widefield images showing A549 WT cells deleted of PARP9, DTX3L or PARP14 in the absence or presence of IFNγ. Cells were stained with Hoechst (Blue), mono-ADPr (AbD43647 IgG-coupled antibody) (Green). (B) Widefield images showing A549 WT cells depleted of PARP9, DTX3L or PARP14, treated with IFNγ (100 ng/mL). Cells were stained with Hoechst (Blue), poly/mono ADPr (poly/mono-ADPr antibody, CST #83732) (Green) and ubiquitin (Santa Cruz, sc-8017) (Magenta). Zoom shows magnified cells as indicated by white box. For all images, scale bar = 20 μm. (C) Quantification of number of ubiquitin foci that also show staining for ADPr from (B). Statistical analysis was determined using a Kruskal Wallis test with Bonferroni correction followed by post-hoc Dunn's test. Asterisks indicate statistical significance compared to control (**$p < 0.01$; ****$p < 0.0001$). Data information: Graph in (C) shows number of foci per individual cell. Between 279 and 466 cells per condition were analysed. The red line indicates the mean foci per cell (siCTRL: 0.266; siPARP9: 1.688; siDTX3L: 0.182; siPARP14: 0.019). Data are representative of a minimum of three (A) or five (B, C) independent replicates. Source data are available online for this figure.

auto-modified PARP14 WWE-ART (Fig. 5A). The PARP9/DTX3L complex, through PARP9 MD1, also hydrolyses modified PARP14 WWE-ART. DTX3L, however, does not possess ADPr-hydrolase activity and fails to remove ADPr from auto-modified PARP14 WWE-ART. These results suggest that while the PARP9/DTX3L complex can suppress PARP14-dependent ADPr through PARP9 macrodomain 1 hydrolytic activity, DTX3L does not modulate PARP14 through hydrolytic activity.

Next, we tested if DTX3L alone or the PARP9/DTX3L complex compromises PARP14 catalytic activity. To reconstitute these reactions in vitro, we generated full-length (FL) PARP14, DTX3L, and the PARP9/DTX3L complex and tested them in a PARP14 auto-ADP-ribosylation assay. Pre-incubating PARP14 full length with either PARP9/DTX3L or DTX3L prior to adding $^{32}$P-NAD$^+$ in the reaction led to a marked reduction of PARP14 auto-ADP-ribosylation (Fig. 5B). Adding a 4x molar excess of either PARP9/DTX3L or DTX3L leads to complete inhibition of PARP14 automodification, which suggests that PARP9/DTX3L and DTX3L inhibit PARP14 catalytic activity in a dose-dependent manner, at least when it comes to the automodification reaction. We also performed this reaction using biotin-labelled NAD$^+$ and achieved the same results (Fig. EV5A). These results suggest that DTX3L could suppress PARP14 ADPr by interacting with PARP14 which compromises its ability to perform ADP-ribosylation, in addition to the hydrolytic removal of ADPr from PARP9 MD1.

According to the AlphaFold-predicted model, the core interaction module between PARP14 and DTX3L is mediated by PARP14 KH8-WWE-ART and DTX3L KH2-3 domains (Fig. 3B) in line with a recently published study which suggested DTX3L KH2-5 domains interact with PARP14 KH8-WWE-ART (Saleh et al, 2024). Since the ART domain of PARP14 is involved in binding DTX3L, it is possible that DTX3L influences PARP14's ADP-ribosylation activity, either by enhancing or inhibiting it. We next modelled the PARP14/DTX3L complex using the core interacting domains, PARP14 KH8-WWE-ART and DTX3L KH2-3 (Figs. 5C and EV5B) with the predicted structure consistent with the DTX3L-PARP14 interactions. The sequence conservation analysis on PARP14 KH8-WWE-ART showed that the DTX3L KH2-3 domains are bound within a groove in PARP14 that is highly conserved across different species (Fig. 5D), indicating the likelihood of the predicted interaction to be true. More specifically, the PARP14 ART domain mainly interacts with DTX3L KH2 domain while PARP14's WWE domain and KH8 domain provide additional supportive interactions. PARP14's KH8 domain, WWE domain and ART domain form a "supermodule", in which KH8 (through its elongated α3 helix) acts as a scaffold to organize the WWE domain and ART domain together (Fig. 5C). Additionally,

we noticed that DTX3L KH2 mainly interacts with PARP14's ART domain, and that DTX3L binding possibly alters the conformation of the donor loop of the PARP14 ART domain and hence influences its catalytic activity.

Based on this model, we then tested if DTX3L can also inhibit PARP14 KH8-WWE-ART activity, which is the core interaction module. Indeed, we observed that PARP14 KH8-WWE-ART automodification was reduced upon pre-incubation with DTX3L FL (Fig. 5E). On the other hand, a shorter fragment of DTX3L RING-DTC (RD) which is not part of the interaction module, inhibits neither PARP14 FL nor PARP14 KH8-WWE-ART automodification (Fig. 5E). We further mutated some residues lining the interface between PARP14 KH8 and DTX3L KH2 (Fig. 5F) and tested if these affected PARP14 automodification. Mutations on KH2 domain of DTX3L (DTX3L FL$^{E292R,Q299R}$) greatly diminished the ability of DTX3L to inhibit PARP14 FL and PARP14 KH8-WWE-ART automodification activity. We also mutated multiple residues on the extended α3 helix on PARP14 KH8 (D1466R, D1468R, E1469R, K1491E, E1525R, D1529R, E1533R). The automodification of PARP14 KH8-WWE-ART bearing these multiple mutations (PARP14 KH8$^{mut}$-WWE-ART) are not inhibited by DTX3L FL and DTX3L FL$^{E292R,Q299R}$ (Fig. 5G). These results suggest that perturbing the predicted DTX3L-PARP14 interaction interface relieves DTX3L inhibitory effect on PARP14 automodification activity. We also observed that both DTX3L FL and DTX3L RD are ADP-ribosylated *in trans* by PARP14 (Figs. 5A,E and EV5). We noted that when the DTX3L-PARP14 interaction interface is perturbed, *trans* ADP-ribosylation of DTX3L becomes more prominent (Fig. 5G). Together, these data suggest that PARP14 activity inhibition is likely achieved via the interaction between DTX3L FL and PARP14 KH8-WWE-ART and also by competition for the modification by other modification substrates (such as DTX3L).

## Discussion

Historically PARP-dependent mono-ADPr has been studied far less than poly-ADPr. This is partly due to the lack of sensitive and specific detection methods. However, signalling by mono-ADPr PARPs, such as PARP14 and PARP7, is important in the regulation of immunity and inflammation (Fehr et al, 2020), although very little is known about the prevalence, targets, or localization of the ADPr signals in physiological conditions. Recently, an improved MAR antibody has been developed (Longarini et al, 2023) and has allowed us to visualize and characterize the endogenous interferon-induced mono-ADPr at the protein level by western-blotting for

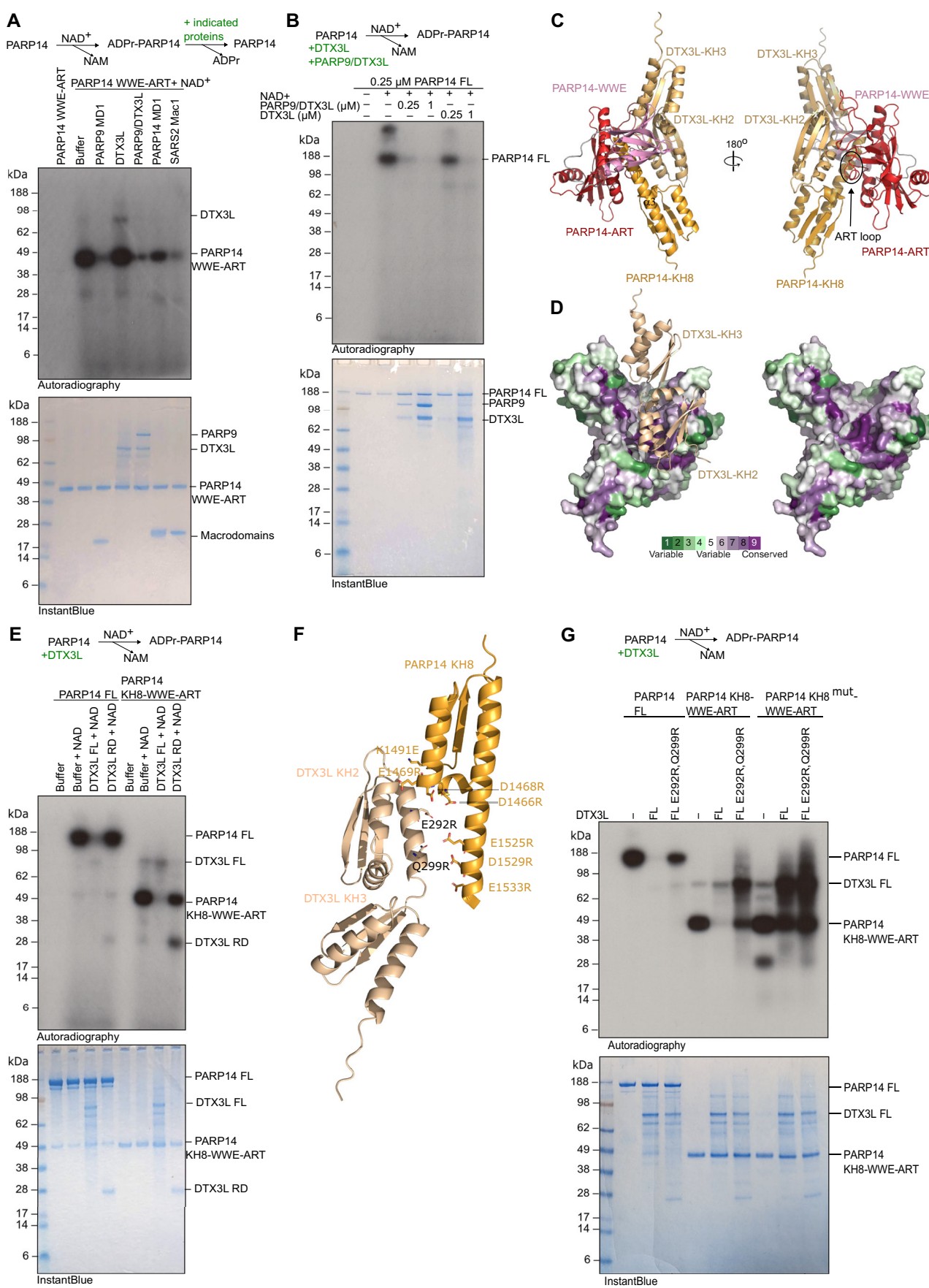

**Figure 5. DTX3L inhibits PARP14 activity in vitro.**

(A) PARP14 WWE-ART automodification reaction with $^{32}$P-NAD$^+$, followed by subsequent addition of the indicated proteins. (B) PARP14 FL auto-ADP-ribosylation reaction performed with $^{32}$P-NAD$^+$ and increasing amount of PARP9/DTX3L or DTX3L. (C) PARP14-DTX3L core interaction module as predicted using AlphaFold. The interaction is mediated by PARP14 KH8-WWE-ART and DTX3L KH2 domain. (D) Surface representation of PARP14 KH8-WWE-ART coloured according to its sequence conservation. Bound DTX3L KH2-KH3 is showed as cartoon. (E) PARP14 FL and PARP14 KH8-WWE-ART auto-ADP-ribosylation reaction performed with $^{32}$P-NAD$^+$ in the presence of DTX3L FL or DTX3L RD. (F) Mutations on the interaction interface between PARP14 KH8 and DTX3L KH2 domain. (G) Auto-ADP-ribosylation reaction of PARP14 FL, PARP14 KH8-WWE-ART, and PARP14 KH8$^{mut}$-WWE-ART performed with $^{32}$P-NAD$^+$ in the presence of DTX3L FL or DTX3L FL mutant. Each experiment has been completed in triplicate.

the first time. We show that the majority of the ADPr signal in cells upon exposure to either IFN or viral nucleic acid mimics is dependent on PARP14, a key PARP mediating coronavirus infection and inflammation responses (Grunewald et al, 2019; Schenkel et al, 2021). Interestingly, we were unable to detect any significant contribution from PARP7, PARP12 or tankyrase PARPs under the same conditions (Figs. 1 and EV1C).

Our data suggest that IFN-induced PARP14 ADPr is regulated in a complex manner, stemming from several different levels. Firstly, we see that PARP14 protein levels are increased after IFN treatment (Figs. 1, 2), in agreement with previous studies showing that PARP14 is one of the major genes whose transcription is induced as a part of IFN response (Fig. 3D) (Moore et al, 2021). However, we also observed that PARP14 protein levels are controlled by its own catalytic activity (Figs. 1 and EV1E,F), mediated by both its hydrolytic macrodomain 1 and its ART domain (Đukić et al, 2023; Schenkel et al, 2021). Indeed, the increased PARP14 protein levels in response to IFN appears to be controlled by both transcriptional upregulation and modulation of its protein degradation/stability via ADPr, the latter possibly dependent on PARP14 automodification.

Additionally, we found that PARP14 is also regulated by the PARP9/DTX3L complex. Firstly, PARP9/DTX3L is important for the IFN-induced upregulation of PARP14 protein levels as we did not see the same robust increase of PARP14 protein levels upon IFNγ stimulation in A549 or RPE1 cells depleted of DTX3L or PARP9 compared to controls (Fig. 3C,G). Importantly, depletion of DTX3L did not reduce transcription of PARP14 after IFN treatment, suggesting instead that the stability of PARP14 protein is altered (Fig. 3D). Furthermore, PARP9/DTX3L also appears to control PARP14 enzymatic ADPr activity. In this respect, we observed that the pattern of protein ADPr dramatically changes upon loss of DTX3L with the ADPr on some proteins increasing and some decreasing in A549 cells. Moreover, the ADPr pattern observed after PARP9 depletion is again different suggesting some non-overlapping or opposing functions of PARP9 and DTX3L. In this respect, we showed that PARP9 can remove PARP14-catalysed ADPr signal, while DTX3L cannot (Figs. 5A and EV3D) (Đukić et al, 2023). In support of partly opposing physiological effects, it was previously reported that PARP9 and PARP14 have antagonistic effects on STAT1 ADP-ribosylation in IFNγ-stimulated macrophage-like cell line THP-1 (Iwata et al, 2016). Our results indicate that there is an intricate network among PARP14, PARP9 and DTX3L in fine-tuning cellular MAR level. When we visualized the cellular localization of ADPr, we observed major changes dependent on depletion of any of PARP14, PARP9 or DTX3L. Upon IFN stimulation, the majority of ADPr signal is found in distinct structures throughout the cytoplasm. Depletion of PARP14 completely abrogated cytoplasmic ADPr while depletion of PARP9/

DTX3L modulated the distribution of ADPr throughout the cell with the formation of smaller ADPr foci. Interestingly, since the effect of DTX3L depletion appears different between immuno-fluorescence and western blotting experiments, this may suggest that different targets of ADPr may be differentially detected by these two techniques, including, for example, protein and nucleic acid ADPr.

Mechanistically, PARP14 and PARP9/DTX3L controlled ADPr signals and their localization can be regulated by multiple mechanisms. Firstly, the hydrolytic activity of PARP9 macrodomain 1 may be responsible for the reversal of some, or all, of PARP14 ADPr in some specific conditions. Indeed, our data shows that PARP9 macrodomain 1 is active against PARP14 model substrates (Figs. 5A and EV3D and (Đukić et al, 2023)). Additionally, we observe that there is a decrease in ADPr of a protein ~50 kDa in size upon depletion or loss of DTX3L or upon overexpression of PARP9 (Figs. 3 and EV3C). Furthermore, our data also suggests PARP9/DTX3L can inhibit or modulate PARP14 ART activity through a physical interaction mediated by DTX3L KH2-KH3 and PARP14 KH8-WWE-ART (Fig. 5B–G). A similar observation that DTX3L reduced PARP14 autocatalytic activity was also reported in a recent study (Saleh et al, 2024). Indeed, our data examining the ADP-ribosylated proteins in interferon-stimulated cells after DTX3L depletion or loss, implicates DTX3L in dampening PARP14 auto-ADPr and modulating *trans* modification on certain targets. Moreover, as PARP9/DTX3L can be ADP-ribosylated (Figs. 5G and EV5A), the complex may also actively compete with PARP14 during automodification, overall reducing PARP14 ADPr.

Current literature has primarily focused on the PARP9/DTX3L complex and how it can regulate both the DNA damage response (Yan et al, 2023; Yan et al, 2009; Yan et al, 2013), as well as host response to pathogens (Zhang et al, 2015). However, it is possible that PARP14/DTX3L may also be a relevant complex in cells. Our models and data suggest that the combined actions of PARP14 and DTX3L produce ADPr-dependent ubiquitination. Specifically, we have been able to visualize the colocalisation of PARP14-dependent ADPr and ubiquitin in cells upon IFN stimulation, which was greatly increased upon depletion of PARP9 (Fig. 4B). We find that the ubiquitination signal is dependent both on PARP14 ADPr and on the E3 ligase DTX3L. Fittingly, we have recently described how the DTX3L is able to ubiquitinate ADP-ribose, resulting in the formation of a composite modification (Zhu et al, 2024; Zhu et al, 2022). It is therefore tempting to speculate that in some instances the colocalisation of ubiquitin and ADPr could represent this double-modification.

Altogether, our approaches have allowed us to begin understanding IFN-induced ADPr and ubiquitylation signalling by PARP14/PARP9/DTX3L. This is of particular importance given

that PARP14 has been reported to have a crucial role in responding to coronavirus infection (Alhammad et al, 2023; Grunewald et al, 2019) and mediating inflammation (Schenkel et al, 2021), while PARP9/DTX3L has been indicated in controlling viral infection and protecting against *Mycobacterium tuberculosis* (Thirunavukkarasu et al, 2023; Zhang et al, 2015). Our data suggests a sophisticated regulation of interferon-stimulated ADPr by these three interferon-stimulated proteins, with PARP14 being a master regulator of ADPr catalysis and PARP9/DTX3L modulating the amount and targets of ADPr. Fittingly, this ADPr signal can be removed by Nsp3 Mac1 which has been shown to be an important virulence factor for SARS-CoV-2, with mutations of Mac1 reducing both viral replication and virulence, which makes it an attractive drug target (Fehr et al, 2018; Grunewald et al, 2019; Kerr et al, 2023; Rack et al, 2020b; Schuller et al, 2023b). Additionally, it is possible that Mac1 homologues from other viruses, such as alphaviruses, will be also efficient in counteracting ADPr by antiviral PARPs (Abraham et al, 2020; Krieg et al, 2023). Finally, it should be noted, that the coronaviral Nsp3 Mac1 and the Mac1-like domains in other viruses can also exert antiviral activity via a non-hydrolytic function through ADP-ribose binding (Abraham et al, 2018; preprint: Kerr et al, 2024).

Altogether, we hope that understanding the signalling process regulated by PARP14/PARP9/DTX3L will help to clarify antiviral response at the molecular level, in addition to understanding how different viruses have adapted to avoid such pathways and how to exploit viral Mac1 proteins as druggable targets (Gahbauer et al, 2023; Leung et al, 2022; Rack et al, 2020b; Roy et al, 2022; Schuller et al, 2023b).

# Methods

## Cell Culture

Human U2OS (ATCC-HTB-96), A549 (ATCC CCL-185), HEK293T (ATCC CRL-3216) and hTERT REP-1 (ATCC CLR-4000) were purchased from American Type Culture Collection (ATCC). All cell lines were maintained in Dulbecco's modified Eagle's medium (DMEM) (Sigma) supplemented with 10% fetal bovine serum (FBS) (Sigma) and penicillin-streptomycin (100 U/mL, GIBCO) and were cultured in a humidified atmosphere at 37 °C with 5% $CO_2$. U2OS and 293T cells were plated 24 h before transfection of plasmids using TransIT-LT1 Transfection Reagent (Mirus Bio) or Polyfect (QIAGEN), respectively, according to the manufactures' instructions. SiRNA-mediated knockdown was performed via reverse transfection in A549, U2OS and RPE-1 cells using Lipofectamine™ RNAiMAX (Invitrogen) according to the manufactures' instructions. The growth media was replaced 24 h after transfection. For transfection of U2OS and A549 cells with nucleic acid mimics, growth media was replaced with DMEM with no FBS or antibiotics before cells were transfected with Poly(I:C) (500 ng/mL) (Tocris), Poly(dA:dT) (500 ng/mL) (InvivoGen), or 5' triphosphate hairpin RNA (500 ng/mL) (InvivoGen) using Lipofectamine™ 3000. Six hours after transfection, the media was replaced with fresh DMEM with FBS and antibiotics. For interferon treatment, growth media was removed from cells and replaced with growth media containing IFNγ (100 ng/mL, Merck) or IFNα (1000 U/mL, Bio-Techne). For PARP14i experiments, cells were pre-treated with PARP14i (0.5 μM, RBN012759, MedChemExpress)

20 min prior to treatment with interferon or nucleic acid mimics. PARP14i was also included in replacement media for nucleic acid mimic experiments. For Fig. 1B, A549 cells were treated with PARP14i (0.5 μM), Veliparib (1 μM, Enzo Life Sciences), PARP7i (0.1 μM, RBN2379, MedChemExpress), TNKSi (1 μM, XAV939, Merck) concurrent with IFNγ treatment. In all experiments, cells were collected for analysis 24 h post interferon, inhibitor or nucleic acid mimic treatment. Cell lines were regularly tested for mycoplasma contamination.

## Knockout cell generation

A549 and U2OS PARP14 knockout cells were generated as previously described (Ran et al, 2013). PARP14 guide 002 was ligated into pSpCas9(BB)-2A-Puro. PARP14 guide 001 and 003 were ligated into pSpCas9n(BB)-2A-Puro. U2OS cells were transfected with pSpCas9(BB)-2A-Puro-PARP14-002 alone or pSpCas9n(BB)-2A-Puro-PARP14-001 and pSpCas9n(BB)-2A-Puro-PARP14-003 together using Lipofectamine® 3000 (Invitrogen) according to the manufacturer's instructions. 72 h post transfection, media was replaced with growth media containing 1 μg/mL puromycin. After 48 h, media was replaced with growth media without puromycin. Cells were isolated by single-cell dilution into 96-well plates. Knockout clones were confirmed by western blotting. sgRNAs used for CRISPR knockouts are described in Appendix Table S1.

U2OS DTX3L knockout cells were generated by nucleofection of ribonucleoprotein (RNP) complexes as previously described (Groslambert et al, 2023), consisting of the Cas9 nuclease pre-loaded with sgRNA as shown in Appendix Table S1. 48 h after nucleofection, single cells were sorted into 96-well plates by FACS. Knockout clones were confirmed by western blotting.

## Western blotting

Western blotting was completed as previously described (Đukić et al, 2023). Briefly, cells were collected 24 h post IFNγ or nucleic acid mimic treatment, or in the case of Fig. EV2 48 h post transfection, washed two times with PBS and lysed with Triton X-100 lysis buffer (50 mM Tris-HCl pH 8.0, 100 mM NaCl, 1% Triton X-100) supplemented with 5 mM $MgCl_2$, 0.1% Benzonase (Sigma), protease and phosphatase inhibitors (Roche), 2 μM Olaparib (Cayman Chemical), 2 μM PARGi PDD00017273 (Sigma) for 30 min at 4 °C. Protein concentrations were determined by Bradford Protein Assay (Bio-Rad) and normalised for equal protein amounts. Proteins were boiled in 1x NuPAGE LDS sample buffer (Invitrogen) with 10–40 mM DTT (Sigma-Aldrich) and resolved on NuPAGE Novex 4%–12% Bis-Tris gels (Invitrogen) in 1x NuPAGE MOPS SDS Running Buffer (Invitrogen) at 150 V. Proteins were transferred onto nitrocellulose membranes (Bio-Rad) using Trans-Blot Turbo Transfer System (Bio-Rad). The membranes were stained with Ponceau S Staining Solution (Thermo Fisher) to check the transfer quality, rinsed with water, and blocked in 5% (w/v) non-fat dried milk in PBS buffer with 0.1% (v/v) Tween 20 (PBST) for 1 h at room temperature. This was followed by overnight incubation with primary antibody as indicated in Appendix Table S2 at 4 °C. The next day, membranes were washed in PBST and incubated with HRP-conjugated antibodies at room temperature for 1 h. Membranes were visualised on Hyperfilm ECL films

(Cytiva) after adding Pierce ECL Western Blotting Substrate (Thermo Fisher). Primary antibodies were used at dilution as described in Appendix Table S2.

## Immunoprecipitation

Dynabeads Protein G (Invitrogen) were washed three times in Triton X-100 buffer (50 mM Tris-HCl pH 8.0, 100 mM NaCl, 1% Triton X-100) with the use of a magnetic separation rack and blocked in 0.5% (v/v) BSA in PBS overnight. The next day, beads were washed three times and incubated with PARP14 or IgG isotype control antibody on an orbital rotator for 2 h at 4 °C. Cells were lysed and protein concentration was measured as described above. 1 mg of cell lysate was added to the antibody-conjugated beads and incubated at 4 °C for 1 h with rotation. Immuno-complexes were washed six times with washing buffer (50 mM Tris-HCl pH 8.0, 300 mM NaCl, 1% Triton X-100). Immunoprecipitated proteins were eluted from the beads using 2×NuPAGE LDS sample buffer (Invitrogen) supplemented with 40 mM DTT (Sigma-Aldrich), boiled for 3 min at 95 °C, and analysed by western blotting.

## RT-qPCR

RT-qPCR was performed as previously described (Đukić et al, 2023). Briefly, A549 cells were treated with siRNA to deplete DTX3L, PARP14 or PARP12 as described above, 72 h prior to harvesting. Cells were treated with IFNγ 24 h prior to harvest. The total RNAs were isolated and purified from A549 cells with RNeasy Plus Micro kit (QIAGEN). Five hundred ng of total RNA was processed for cDNA synthesis with QuantiTect Reverse Transcription Kit according to the manufacturer's instructions. The cDNAs were measured by quantitative real-time PCR using the Rotor-Gene SYBR Green PCR Kit and the Rotor-Gene Q (QIAGEN). Primer pairs for RT-qPCR are shown in Appendix Table S3. The relative gene expression analysis of *PARP14* and *PARP12* was performed using the ddCt method, normalised to hypoxanthine phosphor-ibosyltransferase 1 (*HPRT1*) or glyceraldehyde 3-phosphate dehydrogenase (*GAPDH*) respectively.

## Immunofluorescence

For Fig. 2A–G and Fig. EV2B, A549 cells were plated in 24 well imaging plates (Miltenyi Biotec), allowed to adhere overnight and then treated with IFNγ (100 ng/mL). Cells were washed once in PBS 24 h post IFN treatment before being fixed in ice-cold methanol:acetone (1:1, v/v) at −20 °C for 15 min. Fixative was then removed and cells were washed twice with PBS then blocked in blocking buffer (3% BSA, w/v, in PBS + 0.2% Tween 20). Cells were incubated in primary antibodies diluted in blocking buffer overnight at 4 °C. Primary antibody was removed from wells and the cells were washed three times with PBS + 0.1% Triton. Secondary antibody was diluted 1:500 in blocking buffer with 1.25 μg/mL Hoechst 33342 (Thermo Fisher Scientific) for 1 h at room temperature with gentle rocking. Cells were washed three times with PBS + 0.1% Triton prior to imaging. Cells were imaged on EVOS M7000 widefield microscope equipped with a UPlanSAop 20x/0.75 N.A. objective lens, DAPI, GFP, and Cy™5 EVOS Light Cubes and a CMOS camera. Alternatively, cells were imaged on an

Olympus IX-83 inverted microscope equipped with a Yokogawa SoRa super-resolution spinning-disk head, a PlanApo N 60x/1.4 N.A. SC oil-immersion objective lens and a Prime BSI sCMOS camera. The fluorescence of Hoechst 33342, Alexa Fluor 488 and Alexa Fluor 647 were excited with 405 nm, 488 nm and 647 nm solid state laser respectively and fluorescence detection was achieved with bandpass filters adapted to the fluorophore emission spectra. For Fig. 4 and Fig. EV4, cells were reverse transfected 72 h prior to imaging. For Fig. 2I, cells were plated in an 8-well μ-slide glass bottom chamber slide (ibidi). Cells were transfected with GFP or YFP-Nsp3 Mac1 with Lipofectamine 3000 48 h prior to fixation. Cells were treated with IFNγ 24 h prior to fixation. Images were analysed using CellProfiler (Stirling et al, 2021) to measure mean intensity grey values in the cytoplasm of cells. All experiments were repeated a minimum of 3 times. Images shown are representative of a typical replicate. Antibody dilutions are described in Appendix Table S2.

A custom FIJI script was used to quantify foci containing both ADPr and ubiquitin. The workflow is initiated by segmenting nuclei using Hoechst staining, leveraging the StarDist plugin for accuracy, and delineating the cytoplasm by combining ADP-ribosylation and ubiquitination signals through k-means segmentation. Finally, nuclei are used for marker-controlled watershedding to outline each cytoplasm precisely. Cytoplasmic fluorescent objects were identified and quantified following segmentation based on specific ADP-ribosylation and ubiquitination channel thresholding. The spatial relationships among the segmented objects were elucidated using image arithmetic and binary operations. Image J macros and R scripts have been uploaded together with the source images.

## Plasmids

pDEST-YFP-PARP9 WT, pDEST-YFP-PARP9 G113E, pDEST-YFP-PARP14 WT, pDEST-YFP-PARP14 G832E, FLAG-PARP9, FLAG-Nsp3 Mac1, FLAG-MacroD1 (Đukić et al, 2023) and pEGFP-C1 (Clontech) were generated previously. pDEST-YFP-NSP3 Mac1 was generated by Gateway cloning into pDEST-N-YFP/FRT/TO pcDNA5 from pDonor-Nsp3 Mac1 as previously described (Đukić et al, 2023). pFastBac1 plasmids containing His-PARP14 with a TEV cleavage site, His-DTX3L with a thrombin cleavage site, and MBP-PARP9 with a TEV cleavage site were previously described (Wigle et al, 2020). Mutant His-DTX3L FL$^{E292R,Q299R}$ were generated using QuikChange site-directed mutagenesis kit (Agilent). These plasmids were transformed onto EmBacY cells to prepare bacmid for Sf9 cell infection. Gene fragment for PARP14-KH8$^{mut}$-WWE-ART (D1466R, D1468R, E1469R, K1491E, E1525R, D1529R, E1533R) with *BamHI* and *XhoI* restriction sites was ordered from Invitrogen and cloned into pET28a.

## PARP14 antibody production

The sequence of the 15A6 mouse monoclonal PARP14 antibody (Schenkel et al, 2021) raised against the catalytic domain of human PARP14 was determined from T-cell hybridomas expressing the antibody. The mouse heavy and light chain variable sequences were joined with rabbit IgG constant region sequences and subcloned into the pcDNA3.4 expression vector. The rabbit 15A6 antibody

was expressed in ExpiCHO-S cells and one-step purified on HiTrap MabSelect SuRe affinity columns.

## Protein purification

Purification of these proteins were described previously: His-SARS2 Mac1 (Rack et al, 2020b), His-PARP14 macrodomain 1, His-PARP9 macrodomain 1 (Đukić et al, 2023), His-DTX3L RD (Zhu et al, 2022), His-PARP14 WWE-ART (Munnur et al, 2019), His-PARP14 KH8-WWE-ART (Suskiewicz et al, 2023), His-PARP14 full length (Wigle et al, 2020).

In short, His-tagged proteins (PARP14 macrodomain 1, PARP9 macrodomain 1, DTX3L RD, PARP14 WWE-ART, PARP14 KH8-WWE-ART and PARP14-KH8$^{mut}$-WWE-ART) were expressed in *E. coli* and purified using Ni-NTA resin followed by size exclusion chromatography.

For full-length His-PARP14, His-DTX3L, His-DTX3L/MBP-PARP9, and His-DTX3L$^{E292R, Q299R}$, suspension Sf9 cells were grown in Sf-900 II SFM (Gibco) at 27 °C, 110 rpm, and were infected with the appropriate V1. The cells were harvested by centrifugation and resuspended in lysis buffer (20 mM HEPES pH 7.5, 200 mM NaCl, 0.5 mM TCEP, 5% glycerol, EDTA-free protease inhibitor cocktail (Roche), 0.5% CHAPS, 10 U/mL benzonase). The resuspended cells were lysed using a high-pressure homogeniser and centrifuged at 17,000 rpm at 4 C for 1 h. The clarified lysate was passed through pre-equilibrated HisTrap FF column (Cytiva), followed by washing in 20 mM HEPES pH 7.5, 200 mM NaCl, 20 mM imidazole, 0.5 mM TCEP, 5% glycerol and elution in 20 mM HEPES pH 7.5, 200 mM NaCl, 250 mM imidazole, 0.5 mM TCEP, 5% glycerol. For His-DTX3L/MBP-PARP9, the pooled HisTrap elution fractions were then loaded onto pre-equilibrated amylose resin, washed in 20 mM HEPES pH 7.5, 200 mM NaCl, 0.5 mM TCEP, 5% glycerol, and eluted in 20 mM HEPES pH 7.5, 200 mM NaCl, 10 mM maltose, 0.5 mM TCEP, 5% glycerol. The complex was further purified by size exclusion chromatography (SEC) on a Superdex 200 column (GE Healthcare) in 20 mM HEPES pH 7.5, 200 mM NaCl, 0.5 mM TCEP, 5% glycerol. For His-DTX3L, the HisTrap elution fractions were pooled and further purified with HiTrap SP FF column (Cytiva) and Superdex 200 column (GE Healthcare). For His-PARP14, the HisTrap elution fractions were pooled and further purified with HiTrap Q HP column (Cytiva) and Superdex 200 column (GE Healthcare). Peak fractions containing the proteins were pooled and concentrated using a 30 kDa Viva Spin concentrator (Sartorius). Protein concentration was determined by A280 measurement using Nanodrop and calculated based on molar extinction coefficients. Proteins were aliquoted, snap frozen in liquid $N_2$, and stored at −80 °C.

## PARP14 ADP-ribosylation assay in vitro

For Fig. 5A: 2 µM PARP14 WWE-ART, 40 µM NAD$^+$, 40 nCi/µL $^{32}$P-NAD$^+$ were mixed in reaction buffer (20 mM HEPES pH 7.5, 150 mM NaCl, 1 mM MgCl$_2$, 1 mM DTT) and incubated at 30 °C for 1 h. The automodification reaction was stopped by adding 10 µM PARP14 inhibitor. To initiate hydrolysis, the reactions were supplemented with 2 µM of SARS2 Mac1, PARP14 MD1, PARP9 MD1, His-DTX3L, or His-DTX3L/MBP-PARP9, and incubated at 30 °C for 1 h. Reactions were stopped by adding 3X LDS + 300 mM DTT and boiled at 70 °C for 10 min. Samples were loaded onto 4–12% Bis-Tris gel and run in MES running buffer (Invitrogen) at

180 V. The gel was stained with InstantBlue, destained with 4 washes of milli-Q water, dried gradually using Biorad model 583 gel dryer, and visualised by autoradiography.

For Fig. 5B: His-PARP14 FL (0.25 µM) was pre-incubated with His-DTX3L or His-DTX3L/MBP-PARP9 at equimolar or a 4x molar excess concentration (0.25 µM or 1 µM), on ice for 5 min. NAD$^+$ were added to the reactions to achieve a final concentration of 40 µM NAD$^+$ and 40 nCi/µL $^{32}$P-NAD$^+$, and the tubes were incubated at 30 °C for 1 h. Reactions were stopped, loaded onto SDS-PAGE, and visualised by autoradiography as described above. For reactions using biotin-NAD$^+$ (Fig. EV5), 100 µM NAD$^+$ and 5 µM biotin-NAD$^+$ were used in the reactions. The protein gel was transferred onto nitrocellulose membrane using a TransBlot turbo system (Biorad), blocked with 5% BSA in PBST, stained with Ponceau, detected using Neutravidin DyLight 800, and visualised using LiCor imager.

For Fig. 5E,G: 1 µM of His-PARP14 (FL or KH8-WWE-ART or KH8$^{mut}$-WWE-ART) was pre-incubated with 1 µM His-DTX3L (FL or RD or FL$^{E292R, Q299R}$) on ice for 5 min. NAD$^+$ were added to the reactions to achieve a final concentration of 40 µM NAD$^+$ and 40 nCi/µL $^{32}$P-NAD$^+$, and the tubes were incubated at 30 °C for 1 h. Reactions were stopped, loaded onto SDS-PAGE, and visualised by autoradiography as described above. All assays are repeated three times with the same results.

## Modelling

Molecular model of the PARP14/DTX3L/PARP9 complex was predicted using AlphaFold2. Due to the combined length of the three proteins, it was not possible to predict the trimer in one run. As the first step, the models of full-length proteins were individually predicted. The full-length protein models were split at natural domain boundaries as assessed by PAE plots and the 3D models themselves. Pair-wise complexes were then predicted for all combinations of sub-models, identifying the interfaces. A minimal complex small enough to be predicted as a trimer was then defined from known interfaces as PARP14 (553–1801)/PARP9 (66–830)/ DTX3L (230–515), confirming the same interfaces. The full-length predictions were then aligned to the prediction of the minimal construct to obtain the model of the full-length complex.

Alphafold was run via a local instance of ColabFold v1.2.0 (Mirdita et al, 2022), using Mmseqs2 (Steinegger and Söding, 2017) via a public server (https://api.colabfold.com/ticket/pair) with the default settings (as defined in commit da387b5 of the github repository https://github.com/soedinglab/MMseqs2-App/tree/da387b5) for the MSA generation step. The source search databases were uniref30_2202 (pre-clustered uniref database) and colabfold-d_envdb_202108 (combined metagenomic database). Alphafold-multimer-v2 weights were used in all predictions, with default settings. The relative configuration of domains not participating in the complex interface was not considered structurally or biologically relevant due to inter-model variance and low PAE scores. A single representation of each was instead selected for maximum clarity in the visual presentation.

The PARP14 KH8-WWE-ART (1451–1801) interaction with DTX3L KH2-3 (230–370) was predicted using ColabFold v1.5.2-patch: AlphaFold2 using MMseqs2 (Mirdita et al, 2022) (https://colab.research.google.com/github/sokrypton/ColabFold/blob/main/AlphaFold2.ipynb). ConSurf analysis (https://consurf.tau.ac.il/consurf_index.php) was performed using the PDB output for

PARP14 KH8-WWE-ART (1451–1801) and DTX3L KH2-3 (230–370) generated from AlphaFold2, using standard settings. Structures were visualised in PyMOL (Schrödinger).

## Statistics and reproducibility

For qPCR, box plots in Fig. 3D and Fig. EV1D indicate mean expression levels from three independent biological replicates with error bars indicate average S.D. from three independent replicates. Each biological replicate is an average of quadrupical technical replicates. Statistical significance compared with the control using a Welch's t-test. For image analysis, a minimum of 100 cells per condition and statistical analysis was determined using a Kruskal Wallis test with Bonferroni correction followed by post-hoc Dunn's test. Asterisks indicate statistical significance (ns: not significant, $*p < 0.05$, $**p < 0.01$, $***p < 0.001$, $****p < 0.0001$). Each experiment within this study was completed a minimum of three times with a representative image of the replicates shown.

## Data availability

The source data for microscopy images in Figs. 2A,C,D,G,I, 4A,B, EV2B, EV4A and EV4B have been uploaded to the BioStudies database (McEntyre et al, 2015) (https://www.ebi.ac.uk/biostudies/studies/S-BSST1382). The Biostudies accession number is S-BSST1382.

The source data of this paper are collected in the following database record: biostudies:S-SCDT-10_1038-S44318-024-00126-0.

## Peer review information

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

## Acknowledgements

We would like to thank Ivan Matic for his kind gift of IgG- and HRP-conjugated anti-mono-ADPr antibodies. We thank the Soeding lab for generously providing public access to their MMSeqs2 prediction server. We also thank the Alan Wainman and the Dunn School Bioimaging Facility for expert advice and access to the microscope. We would like to thank Joey Riepsaame and the Genome Engineering Oxford (GEO) Facility for help making the DTX3L knockout cells. The work in Ivan Ahel's lab is supported by the following grants: Biotechnology and Biological Sciences Research Council (BB/R007195/1 and BB/W016613/1), Wellcome Trust (210634), Oxford University Challenge Seed Fund (USCF 456), Ovarian Cancer Research Alliance (813369). Ivan Ahel and Sumana Sanyal are jointly supported by Wellcome Trust (223107). The work in Dragana Ahel's lab is supported by the Edward Penley Abraham Research Fund. The Huet lab receives funding from the Agence Nationale de la Recherche (ANR-22-CE12-0039 AROSE), the Fondation ARC pour la recherche sur le cancer (ARCPJA2022060005190) and the Ligue Contre le Cancer (CD53).

## Author contributions

**Pulak Kar**: Investigation; Writing—original draft. **Chatrin Chatrin**: Investigation; Visualization; Methodology; Writing—original draft; Writing—review and editing. **Nina Đukić**: Data curation; Investigation; Visualization; Writing—review and editing. **Osamu Suyari**: Investigation. **Marion Schuller**: Investigation. **Kang Zhu**: Investigation. **Evgeniia Prokhorova**: Investigation. **Nicolas Bigot**: Formal analysis; Investigation. **Juraj Ahel**: Formal analysis; Investigation. **Jonas Damgaard Elsborg**: Investigation. **Michael L Nielsen**: Supervision. **Tim Clausen**: Supervision. **Sébastien Huet**: Formal analysis; Supervision. **Mario Niepel**: Resources. **Sumana Sanyal**: Supervision; Funding acquisition. **Dragana Ahel**: Supervision; Funding acquisition. **Rebecca Smith**: Data curation; Formal analysis; Supervision; Investigation; Visualization; Methodology; Writing—original draft; Writing—review and editing. **Ivan Ahel**: Conceptualization; Supervision; Funding acquisition; Writing—original draft; Project administration; Writing—review and editing.

Source data underlying figure panels in this paper may have individual authorship assigned. Where available, figure panel/source data authorship is listed in the following database record: biostudies:S-SCDT-10_1038-S44318-024-00126-0.

## Disclosure and competing interests statement

MN is an employee and shareholder of Ribon Therapeutics. EP is an employee of Vertex Pharmaceuticals and may own stock or stock options in that company. All other authors declare that they have no competing interests.

# Expanded View Figures

**Figure EV1.  PARP14 is auto-ADPr upon immune stimulation and this regulates its stability.**

(**A**) A549 cells were treated with 0.5 μM PARP14i (RBN012759) and/or with IFNɣ (100 ng/mL). Cell lysates were examined by western blot with the indicated antibodies. (**B**) A549 cells were treated with 0.5 μM PARP14i (RBN012759) and/or with IFNɣ (100 ng/mL). Cell lysates were subjected to immunoprecipitation with Protein G beads conjugated with indicated antibodies. The ADPr signal was analysed using western blotting. (**C**) A549 cells depleted or not of PARP14 or PARP12 were treated with IFNɣ (100 ng/mL) where indicated. Cell lysates were examined by western blot with the indicated antibodies. (**D**) Relative gene expression of *PARP12* for (**C**) in unstimulated and IFNɣ (100 ng/mL) stimulated A549 cells was determined by RT-qPCR. Gene expression levels were normalized to the expression of *GAPDH*. Error bars indicate average S.D. from three independent replicates. Asterisks indicate statistical significance compared with the control, as determined by Welch's t-test (ns: not significant, \*\*$p < 0.01$, \*\*\*$p < 0.001$ Two-tailed $P$ value, siCTRL vs siPARP12 -IFNɣ $p = 0.002$, siCTRL vs siPARP12 +IFNɣ $p = 0.0006$). (**E**) U2OS cells were transfected with the indicated plasmids in the presence or absence of 0.5 μM PARP14i or expression of FLAG-SARS2 Mac1. The levels of YFP-tagged PARP14 in cell lysates were assessed using an anti-GFP antibody. (**F**) U2OS cells were transfected with indicated plasmids in the presence or absence of PARP14 inhibitor (0.5 μM). Cells were lysed 12, 24, 32 or 48 h after transfection and analysed by western blot using the indicated antibodies. For all blots, pSTAT1 antibody was used as a positive control for stimulation of immune response. GAPDH or tubulin were used as a loading control. Source data are available online for this figure.

▶

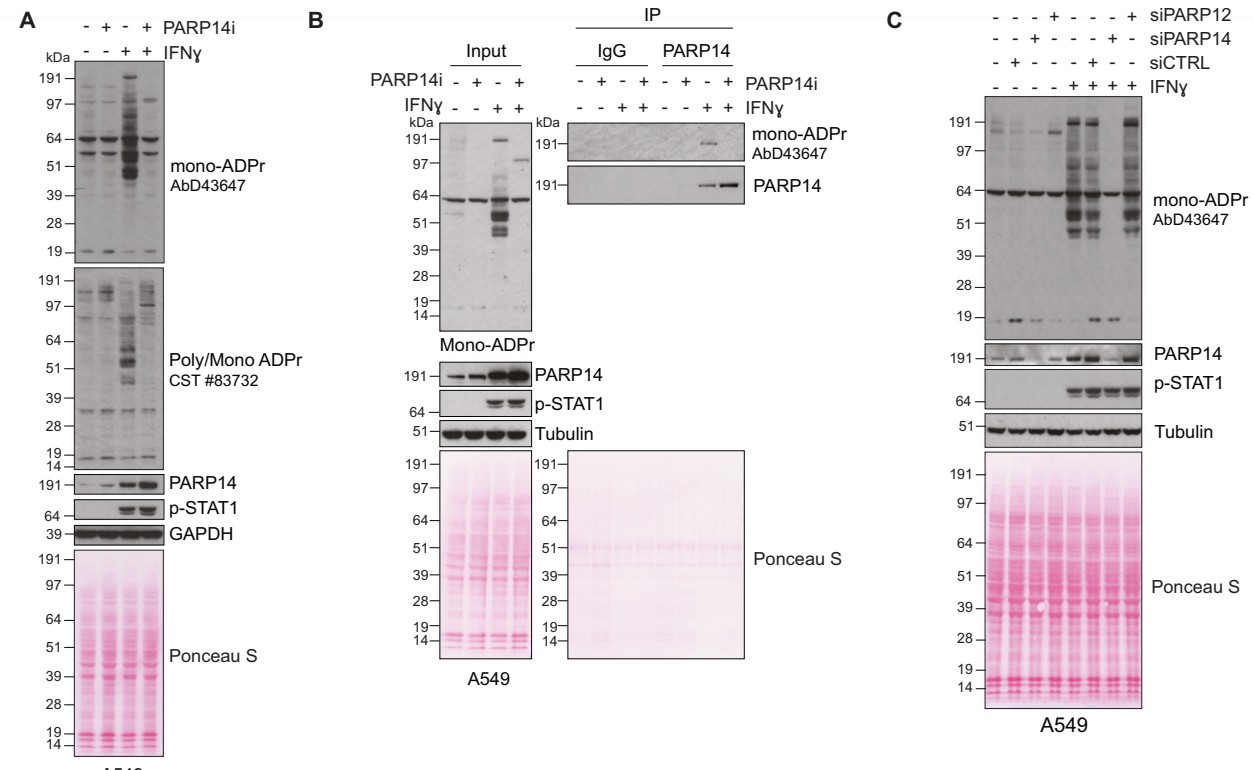

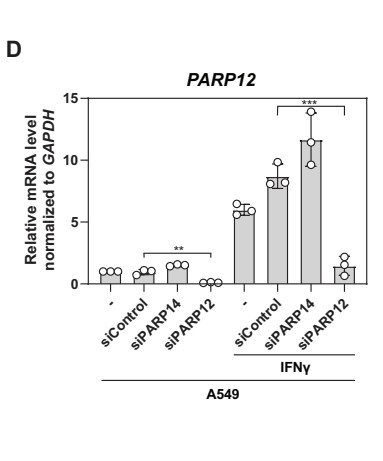

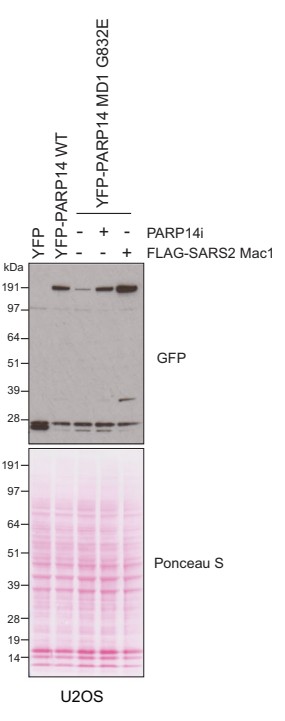

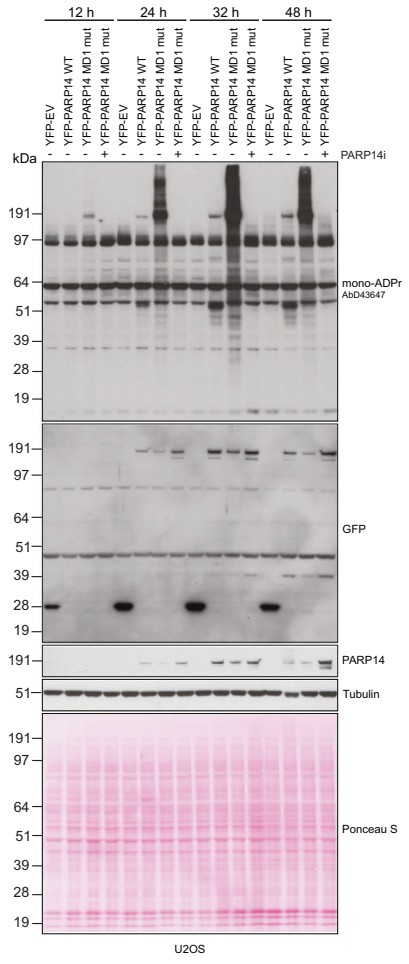

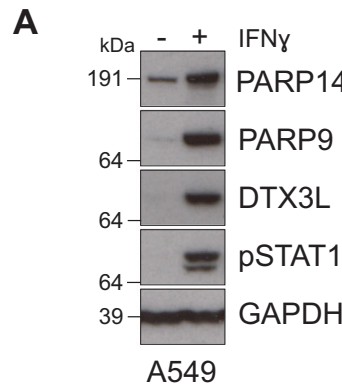

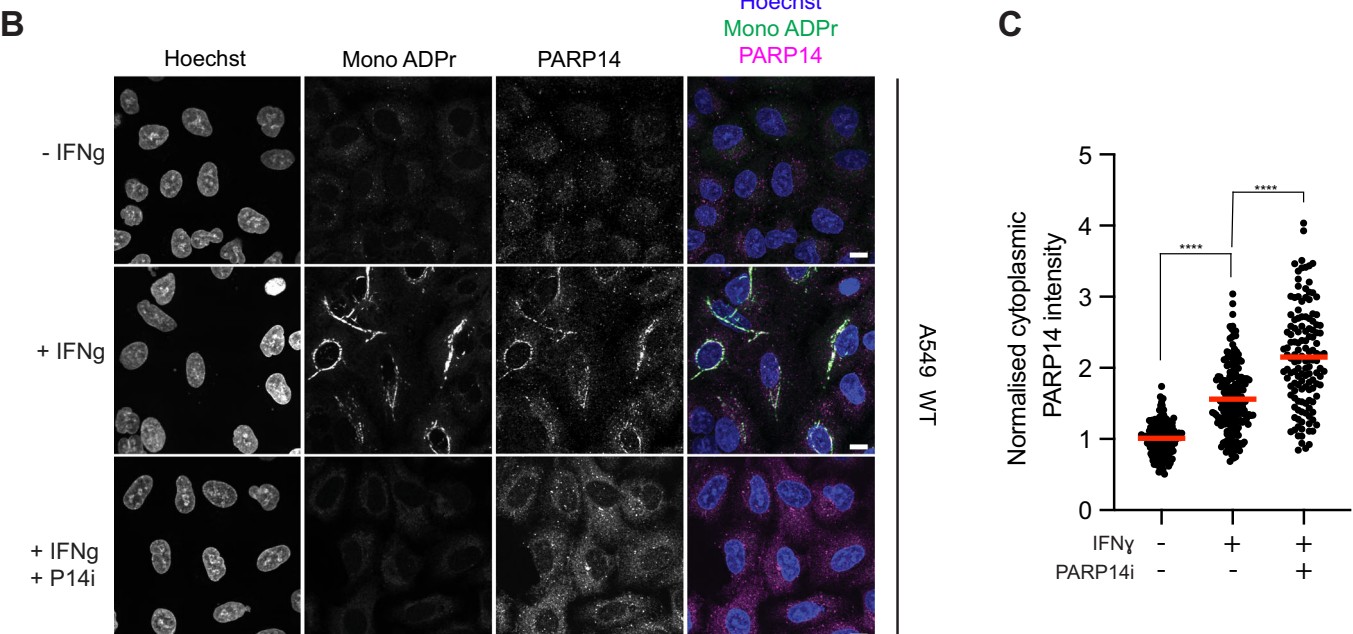

**Figure EV2.  PARP14, PARP9 and DTX3L increase after IFNγ stimulation.**

(**A**) A549 cells were treated or not with IFNγ (100 ng/mL). Cell lysates were analysed by western blot using the indicated antibodies. GAPDH was used as a loading control. pSTAT1 indicates induction of the interferon response. (**B**) Confocal images showing A549 cells untreated or treated with IFNγ (100 ng/mL) in the presence or absence of PARP14i (0.5 μM). Cells were stained with Hoechst (Blue), mono-ADPr (AbD43647 IgG-coupled) (Green) and PARP14 (Abcam, ab224352) (Magenta). Scale bars, 20 μm. (**C**) Normalised intensity of cytoplasmic PARP14 signal from (**B**). Fluorescence intensity was normalised to WT cells in the absence of IFNγ or PARP14i. The red line indicates mean intensity. Statistical analysis was determined using a Kruskal Wallis test with Bonferroni correction followed by post hoc Dunn's test. Asterisks indicate statistical significance (****$p < 0.0001$). Data are representative of two independent replicates. Source data are available online for this figure.

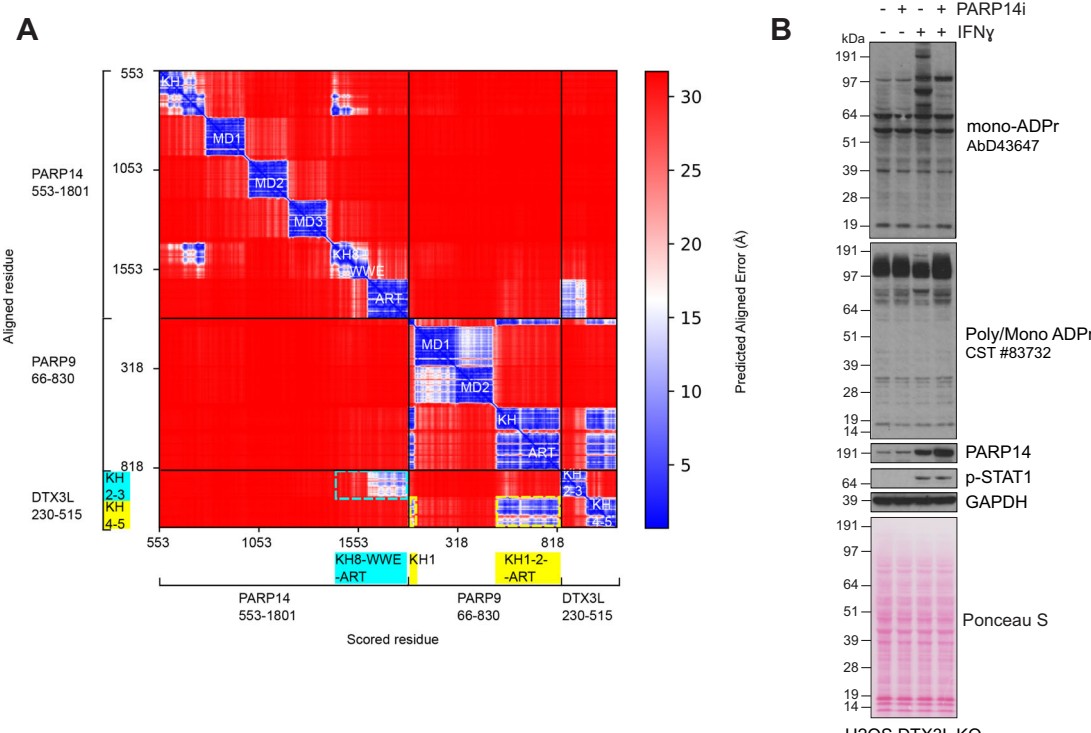

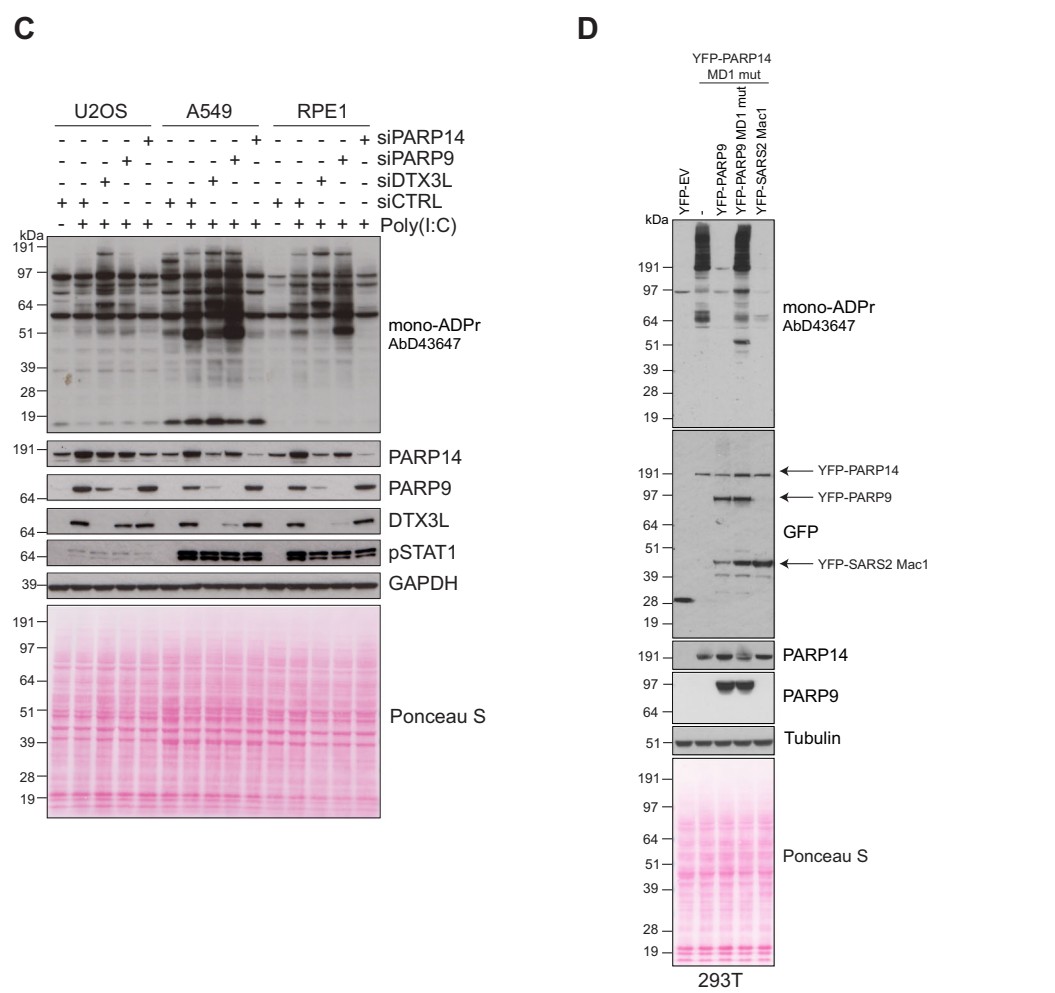

**Figure EV3.   PARP9/DTX3L regulate PARP14-dependent ADPr.**

(A) Predicted aligned error (PAE) plot of PARP14 (553–1801)/DTX3L (230–515)/PARP9 (66–830) model. Inter-domain interaction between PARP14 KH8-WWE-ART and DTX3L KH2-3 is indicated in cyan dashed box. Inter-domain interaction between PARP9 KH-ART and DTX3L KH4-5 is indicated in yellow dashed box. There are also notable intra-domain contacts within each protein: PARP14 KH-WWE, PARP9 KH-ART, PARP9 tandem macrodomains, DTX3L KH2-3, and DTX3L KH4-5. The relative orientations of other domains with high PAE score are uncertain. (B) U2OS DTX3L KO cells were treated with 0.5 µM PARP14i and/or 100 ng/mL IFNγ. Cell lysates were examined by western blot with the indicated antibodies. (C) U2OS, A549 and RPE1 cells depleted or not of DTX3L, PARP9 or PARP14 were treated with poly(I:C). Cell lysates were examined by western blot with the indicated antibodies. For all blots, pSTAT1 antibody was used as a positive control for stimulation of immune response. (D) 293T cells were transfected with YFP-tagged PARP14 MD1 mutant alone or co-transfected with indicated plasmids. Cell lysates were collected 24 h after transfection and examined with the western blot using indicated antibodies. GAPDH or tubulin were used as a loading control in all blots. Source data are available online for this figure.

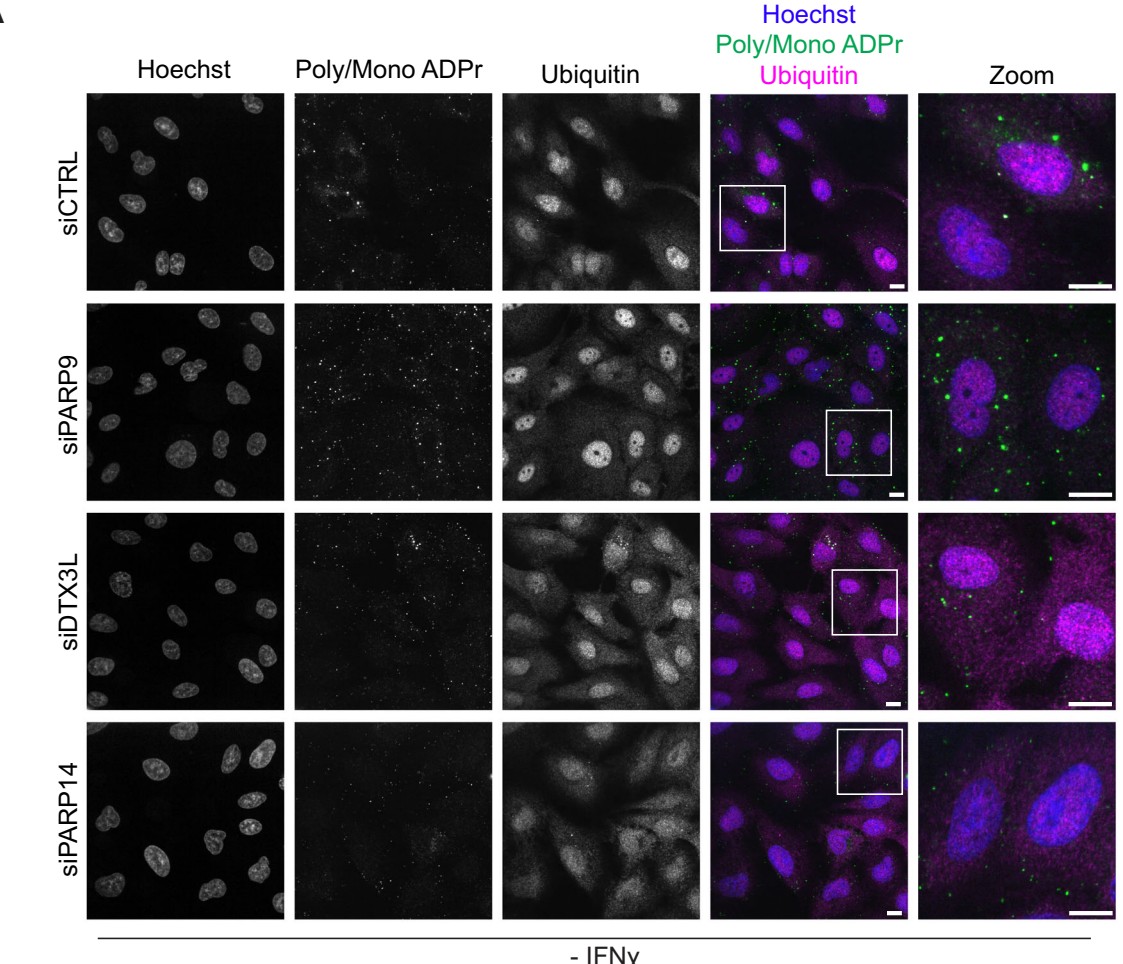

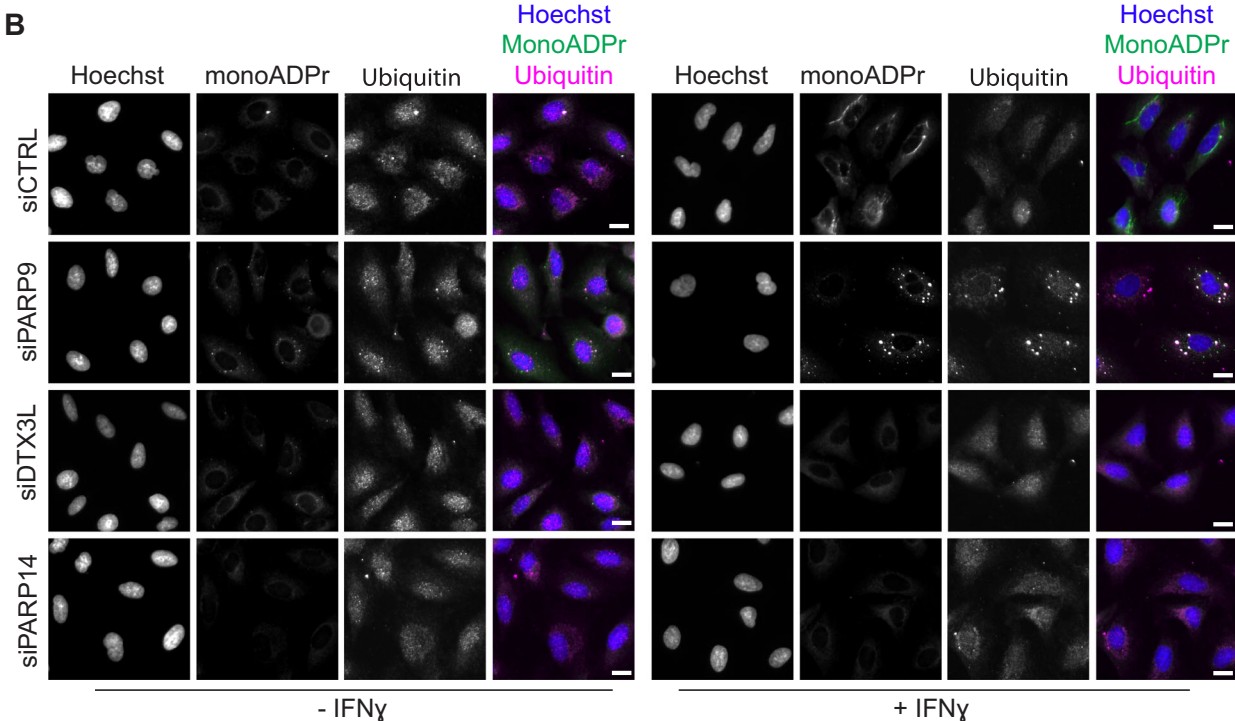

◀ **Figure EV4. PARP14, PARP9 and DTX3L regulate ADPr and ubiquitin foci formation.**

(A) Confocal images showing A549 WT cells deleted of PARP9, DTX3L or PARP14 in the absence of IFNγ. Cells were stained with Hoechst (Blue), ADPr (poly/mono-ADPr antibody, CST #83732) (Green) and ubiquitin (Magenta). (B) Widefield images showing A549 WT cells depleted of PARP9, DTX3L or PARP14, untreated or IFNγ-treated (100 ng/mL). Cells were stained with Hoechst (Blue), mono-ADPr (AbD43647 IgG-coupled antibody) (Green) and ubiquitin (Abcam, ab134953) (Magenta). For all images, scale bar = 20 µm. Data information: Data are representative of a minimum of five (A) or three (B) independent replicates. Source data are available online for this figure.

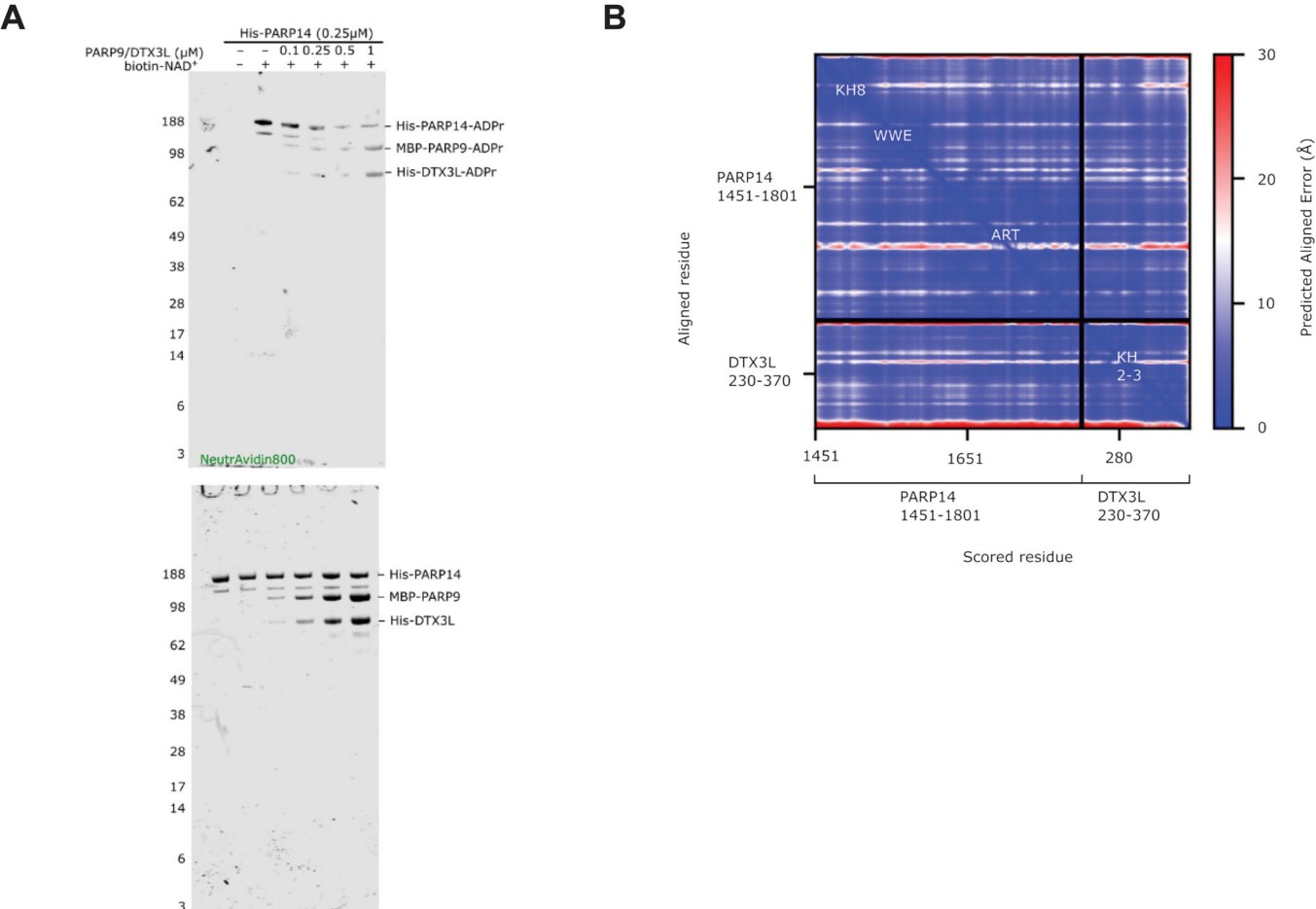

**Figure EV5.  DTX3L inhibits PARP14 activity in vitro.**

(A) PARP14 FL auto-ADP-ribosylation reaction performed with biotin-NAD+ and increasing amount of PARP9/DTX3L. (B) Predicted aligned error plot of PARP14 KH8-WWE-ART (residue 1451–1801) and DTX3L KH2-3 (residue 230–370) model.

