## [Peer Review File · The EMBO Journal]

PARP14 and PARP9/DTX3L regulate interferon-induced ADP-ribosylation

Pulak Kar, Chatrin Chatrin, Nina Đukic, Osamu Suyari, Marion Schuller, Kang Zhu, Evgeniia Prokhorova, Nicolas Bigot, Juraj Ahel, Jonas Damgaard Elsborg, Michael L Nielsen, Tim Clausen, Sébastien Huet, Mario Niepel, Sumana Sanyal, Dragana Ahel, Rebecca Smith, and Ivan Ahel

Corresponding author(s): Ivan Ahel (ivan.ahel@path.ox.ac.uk) , Rebecca Smith (rebecca.smith@path.ox.ac.uk)

Review Timeline:

Submission Date:	7th Oct 23
Pre-decision consultation:	24th Nov 23
Authors' Revision Plan:	7th Dec 23
Editorial Decision:	15th Dec 23
Revision Received:	4th Apr 24
Editorial Decision:	30th Apr 24
Revision Received:	1st May 24
Accepted:	8th May 24

Editor: Hartmut Vodermaier

Transaction Report:

Dear Ivan,

Thank you again for your patience during the peer review of your manuscript, which has now concluded. As you will see from the comments below, the referees appreciate the general importance of the topic and the potential interest of your findings. However, especially referees 2 and 3 also raise a number of issues, related to validation/characterization of reagents, conclusiveness/interpretation of particular experiments, and the preliminary nature of some aspects of the work. Since it is not clear if and how these issues could be adequately addressed during a regular round of major revision, I would like to give you a chance to consider the reports and to provide a tentative point-by-point response, prior to taking a final decision in this case. I would be especially interested to hear how the concerns about the preliminary AF2 modeling and ubiquitin data might be experimentally addressed. Based on such a proposal and preliminary point-by-point response to the referees' comments, I could then determine whether a (scooping-protected) major revision for The EMBO Journal would seem realistic, or whether a less substantively revised version might at least be suitable for one of our sister journals. I'd also be happy to talk through such a revision proposal with you if needed.

Looking forward to hearing from you,

Best regards,

Hartmut

Referee #1 (Report for Author)

The manuscript by Kar et al. establishes PARP14 as the primary member of the PARP family accountable for IFN γ (and other immune activators)-stimulated cellular ADP-ribosylation. The researchers also present supporting evidence suggesting that PARP9/DTX3L regulates PARP14 levels in cells by directly regulating PARP14's catalytic activity. They use alphafold modeling to suggest how PARP9/DTX3L could bind to PARP14 and regulate its catalytic activity. They then show in vitro that PARP9/DTX3L can inhibit the catalytic activity of PARP14. This manuscript is well-crafted and provides keen insight into IFN γ -stimulated mono-ADP-ribosylation and the regulation of PARP14 catalytic activity. The increasing interest in PARP14 as a therapeutic target makes this study timely and should be of broad interest to the EMBO readership. Some minor points should be addressed prior to publication:

1. Fig. 1A: the PARP14i + IFN γ WT condition seems missing.
2. Fig. 1 title: states that PARP14 is the major ART activated in response to immune stimuli. This could be rephrased to say that PARP14 catalytic activity is required for ADP-ribosylation induced by IFN γ and other immune activators. It is hard to say if these stimuli "activate" PARP14 as they also increase PARP14 protein levels.
3. Fig EV1: the ADPr Western blot should be shown.
4. Fig. 2A: Does the ADPr signal they observe co-localize with mitochondria? The authors could do a co-staining experiment with mitochondrial markers to address this.
5. Fig. 4B: the authors should quantify the co-localization of ubiquitin with ADPr.
6. Fig. 4: can the authors speculate why the ADPr pattern changes so dramatically between the control and siPARP9 conditions (appears to change from a more clustered, perinuclear localization to a more punctate localization).

Referee #2 (Report for Author)

Kar et al. show that addition of IFN-gamma to some cells (A549 and RPE1, but not U2OS) results in increased cellular ADP-ribosylation (ADPr) as detected by a new anti-ADPr antibody. They show that this increased ADPr signal is dependent on the catalytic activity of PARP14 and can be reduced by overexpression of a macrodomain (Mac1) from SARS-CoV-2. They further show that PARP14 activity is regulated by auto-ADP-ribosylation as well as interactions with the PARP9/DTX3L complex. Finally, they provide microscopy data that cytoplasmic ubiquitin co-localizes and is dependent on IFN-induced cytoplasmic ADPr.

There are several interesting aspects to this manuscript. The authors provide clear and convincing data on the requirement for PARP14 in the IFN-induced increase in ADPr signal. In addition, they clearly show that PARP14 activity is regulated by both PARP14 ADP-ribosylation as well as the presence of PARP9/DTX3L. Both of these pieces of data further support the model that has been emerging from their work and others that PARP14, PARP9, and DTX3L are important regulators of the IFN-gamma response. However, in several places, the authors overinterpret their data, drawing conclusions that are not supported by the data, which weakens enthusiasm for the paper. Detailed comments are below:

Major concerns:

1) Many of the results are dependent on one specific antibody and it would appear that different antibodies give slightly different results in some of these assays (both in this manuscript and the co-submitted manuscript by Ribeiro et al). Can the authors compare antibodies in both western blot and fluorescence assays to be fully transparent about when there is an IFN-gamma-induced increase in ADPr? Detection of this signal in an antibody-specific

way does not discount its existence, but it would be useful for future studies for the authors to be clear about when this can be observed and when it can not and even provide some speculation about the reasons for those differences.

2) The authors do not formally show that the ADPr signal that they detect is directly catalyzed by PARP14. Given all of the cross-regulation of PARPs, and the dependence of these enzymes on each other (some of which is nicely shown in this paper), it would be more appropriate to be interpret these results as indicating that PARP14 activity is required for this increase in ADPr signal rather than that is the "main enzyme responsible for this modification".

3) The authors go into great molecular detail about a predicted AlphaFold model of a complex between PARP14, DTX3L, and PARP9. However, they do not validate this model in any way. The atomic model they describe should provide predictions for specific mutations that can be made to disrupt the interaction and thereby validate their predicted AlphaFold model (e.g. their detailed Fig. 5C and 5D). Simply removing the N-terminal half to 2/3 of DTX3L and showing that this interaction is lost is not sufficient to validate that model, and does not justify overly interpreting a predicted AlphaFold model.

4) In many cases, the authors are careful to not overinterpret the data on SARS-CoV-2 Mac1. However, this does not extend to the abstract. Their data do not provide "the mechanism by which Mac1 counteracts the activity of antiviral PARPs". Their data show that overexpressed Mac1 is able to reverse ADP-ribosylation, which is a result that has been shown repeatedly on a variety of ADP-ribosylated substrates. Thus, removal of the IFN-induced ADPr is one potential mechanism for Mac1 function, but by far not the only potential mechanism. Do other overexpressed macrodomains not show the same decrease in ADPr signal? Does infection with a Mac1-active, but not Mac1-inactive virus, cause this same decrease in ADPr signal? These types of experiments would get the authors closer to that kind of mechanistic interpretation. Notably, a recent paper (<https://pubmed.ncbi.nlm.nih.gov/37695054/>) showed that Mac1 mutant virus is inhibited in vivo in a PARP12-dependent manner. Based on this paper, which the authors should cite, as well as many other potential mechanisms that have been proposed for Mac1 activity, the authors should be much more cautious in their interpretation of their macrodomain data.

5) The data on ubiquitin in Figs. 4B and EV4 are very preliminary. It is difficult to draw any meaningful conclusions from single images that contain <10 cells, only one or two are magnified sufficiently to see any foci or colocalization. Using those images alone, one is unable to conclude that the authors have presented "the first visualisation of ADPr-dependent ubiquitylation in the IFN response". Additional images and quantification are required if the authors want to conclude this.

Other concerns:

1) The antibody used for PARP14 in Figure 2 does not appear to be entirely specific for PARP14 given how much signal remains in the PARP14 KO cells. Do the authors have another way to support the claim that PARP14 co-localizes with ADPr signal in those cells?

Referee #3 (Report for Author)

The study from the Ahel group addresses the contribution of PARP14 to IFN-induced ADP-ribosylation in cell lines, and possible interplay with DTX3L/PARP9. There are two approaches that led to the primary observations described in the manuscript. First, a recently developed antibody probe for mono-ADP-ribosylation was used in blotting experiments to identify (by molecular weight) several polypeptides that undergo PARP14 activity-dependent ADP-ribosylation in response to IFN treatment, which has been shown on other studies to induce PARP14 expression. The same antibody was used in fluorescence microscopy experiments to show the ADP-ribose signal generated by PARP14 localizes in a perinuclear zone that in some cells is visualized as small foci. Second, biochemical data is presented that shows PARP14 activity can be regulated by PARP9 via a hydrolysis-active macrodomain in the latter. In this manner, PARP9 bound to PARP14, or DTX3L/PARP9 bound to PARP14, is envisioned to modulate PARP14 output. Because PARP14 and DTX3L/PARP9 are all induced by IFN signaling, the data suggest that PARP14 output may be titrated by an "opposing" level of DTX3L/PARP9, though other scenarios (including effects on the levels of specific substrates) can be envisioned. Overall the experiments in the manuscript are well executed but some additional data will help solidify the molecular mechanism and reveal the importance of the biochemical observations with regard to IFN signaling.

Figure 1, 2: It is a surprising that despite two figures with very nice quality blotting and IF are shown with the new antibody, but we are still left with no knowledge as to the identity of the PARP14 substrates. The authors infer that PARP14 is a major substrate based on MW, and this should be verified by IP and blotting. Given that different labs even within the field are using different reagents for detection of ADP-ribose, it would be helpful if the specific antibody name was used in the figure panels throughout the paper, or minimally in the figure legends. Also, p-STAT1 blotting is shown but there was no comment in the results on the effect of PARP14i (or lack thereof) as the data especially in Figure 1C indicates PARP14 has no apparent role in JAK-STAT signaling - at least through STAT1. Consistent with this view, Figure 3C shows that neither PARP14 KO nor DTX3L siRNA (and the associated loss of PARP9) has no obvious effect on STAT1 and p-STAT1 induction by IFN. The former also does not affect IFN induction of DTX3L/PARP9. These data would seem to rule out a PARP14 contribution to the core signaling and transcription events that follow IFN treatment. Please comment. Also, statistical tests should be applied to the image quantitation, which is shown.

Figure 3: The authors should clarify if there are cell-line or technical differences in terms of DTX3L/PARP9 influence on PARP14 expression (e.g. siRNA vs KO's). Also, with the data spread (Fig. 3D) it is hard to tell if siDTX3L affects PARP14 message levels +IFN in A549 cells.

Figure 4: It may not be obvious how to reconcile the images with the treatment conditions, particularly a comparison of panel A vs panel B. PARP9 depletion appears to have minimal or no effect on IFN-induced mono-ADP-ribosylation in either panel, yet PARP9 MD modulation of PARP14 catalytic activity is the model put forth in the manuscript.

Figure 5: Data are presented to show the MD of PARP9 can remove ADP-ribose from PARP14. Unless I am misunderstanding the experimental design, using PARP14 auto-modification state as the readout for PARP14 activity in this case may not be sufficient to draw conclusions regarding an effect of PARP9 on PARP14 activity. Since the treatment condition (supplementing the second phase of the reaction with a MD, which hydrolyzes ADP-ribose from auto-modified PARP14) would release ADP-ribose even from fully-active PARP14, it is not obvious how this event necessarily modulates PARP14 activity - unless the specific auto-ADP-ribosylated sites in PARP14 are mapped, mutated, and shown to be necessary for catalytic activity. It seems the data that points to a role for DTX3L/PARP9 in modulating PARP14 is Figure EV3B, where siRNA was used against PARP9 and mono-ADP-ribosylation induced by Poly(I:C) followed by blotting. In U2OS cells, PARP9 KD had little or no effect on PARP14 mono-ADP-ribosylation (lanes 2, 4). In A549 cells, PARP9 KD probably had some effect on the intensity of some bands labeled by PARP14 (lanes 7, 9). The most convincing effect was on RPE1 cells, where PARP9 KD increased the intensity of the ~51 kDa band (lanes 12, 14). There is a slight concern about using the ADPr readout on blots (without knowing if the substrate protein levels are changing) since DTX3L/PARP9 depletion could have effects independent of PARP14 that are manifest as ADP-ribosylation differences but actually involve changes in protein levels via E3-mediated degradation, or loss thereof. This gets back to the point that if knowing the specific substrates labeled by PARP14 (such as the 51 kDa band) would position the group for a more robust test of the models.

Lastly, to extend and validate these findings in cells, the authors should be able to reconstitute PARP9 KO cells with WT and PARP MD1 hydrolysis mutant constructs and determine the effect on PARP14 auto-modification and PARP14 activity towards the substrates shown by blotting in Figure 1. This will provide a cell-based test of the model for DTX3L/PARP9 regulation of PARP14, and it will also shed light on the relative contributions of simple binding to PARP14 versus removal of ADP-ribose from PARP14.

Minor comment on Figure EV1: PARP14 protein stability cannot be assessed from a single timepoint.

Referee #1 (Report for Author)

The manuscript by Kar et al. establishes PARP14 as the primary member of the PARP family accountable for IFN γ (and other immune activators)-stimulated cellular ADP-ribosylation. The researchers also present supporting evidence suggesting that PARP9/DTX3L regulates PARP14 levels in cells by directly regulating PARP14's catalytic activity. They use alphaFold modeling to suggest how PARP9/DTX3L could bind to PARP14 and regulate its catalytic activity. They then show in vitro that PARP9/DTX3L can inhibit the catalytic activity of PARP14. This manuscript is well-crafted and provides keen insight into IFN γ -stimulated mono-ADP-ribosylation and the regulation of PARP14 catalytic activity. The increasing interest in PARP14 as a therapeutic target makes this study timely and should be of broad interest to the EMBO readership. Some minor points should be addressed prior to publication:

1. Fig. 1A: the PARP14i + IFN γ WT condition seems missing.
We hope the reviewer can appreciate that these conditions have been included in both Figure 1B and 1C. In addition, when we address comments from reviewer 2 regarding the use of different antibodies to detect ADPr, we will include also this condition.
2. Fig. 1 title: states that PARP14 is the major ART activated in response to immune stimuli. This could be rephrased to say that PARP14 catalytic activity is required for ADP-ribosylation induced by IFN γ and other immune activators. It is hard to say if these stimuli "activate" PARP14 as they also increase PARP14 protein levels.
We shall address this comment within the text to clarify the comments.
3. Fig EV1: the ADPr Western blot should be shown.
We can include such data. We will also likely combine this with a comment from reviewer 3 who has requested additional timepoints to show that loss of the PARP14 MD1 hydrolytic activity reduces PARP14 stability.
4. Fig. 2A: Does the ADPr signal they observe co-localize with mitochondria? The authors could do a co-staining experiment with mitochondrial markers to address this.
We agree that the staining appears to resemble a network of mitochondria. We have previously assessed mitochondria and IFN γ induced ADPr and not seen colocalization between the two signals.
5. Fig. 4B: the authors should quantify the co-localization of ubiquitin with ADPr.
This will be addressed.
6. Fig. 4: can the authors speculate why the ADPr pattern changes so dramatically between the control and siPARP9 conditions (appears to change from a more clustered, perinuclear localization to a more punctate localization).
We are currently working through different hypotheses to address this comment. However, we can speculate that because PARP9 MD1 is able to hydrolyse PARP14 ADPr (Dukic et al, Sci Adv 2023), the loss of PARP9 (and its hydrolytic MD1) might preserve PARP14 ADPr, which may lead to different localisation pattern. Indeed, we also see different patterns of YFP-PARP14 WT and MD1 mutant (Dukic et al, Sci Adv 2023). Moreover, as ADPr has been shown to be a catalyst of phase separation, one could speculate that the increased ADPr

seen on western blots with the PARP14 MD1 mutant could promote the accumulation of ADPr or ADP-ribosylated proteins into these more punctate foci. If this is the case, it could suggest that these structures may not stain equally well with antibody during IF and could thus account for the discrepancies between our immunofluorescence data and our western blots.

We could also speculate that if PARP14/DTX3L/PARP9 complex bind to other partners, and that these interactions could be perturbed upon PARP9 loss and thus we have a change in structure when the large complex is disrupted.

Referee #2 (Report for Author)

Kar et al. show that addition of IFN-gamma to some cells (A549 and RPE1, but not U2OS) results in increased cellular ADP-ribosylation (ADPr) as detected by a new anti-ADPr antibody. They show that this increased ADPr signal is dependent on the catalytic activity of PARP14 and can be reduced by overexpression of a macrodomain (Mac1) from SARS-CoV-2. They further show that PARP14 activity is regulated by auto-ADP-ribosylation as well as interactions with the PARP9/DTX3L complex. Finally, they provide microscopy data that cytoplasmic ubiquitin co-localizes and is dependent on IFN-induced cytoplasmic ADPr.

There are several interesting aspects to this manuscript. The authors provide clear and convincing data on the requirement for PARP14 in the IFN-induced increase in ADPr signal. In addition, they clearly show that PARP14 activity is regulated by both PARP14 ADP-ribosylation as well as the presence of PARP9/DTX3L. Both of these pieces of data further support the model that has been emerging from their work and others that PARP14, PARP9, and DTX3L are important regulators of the IFN-gamma response. However, in several places, the authors overinterpret their data, drawing conclusions that are not supported by the data, which weakens enthusiasm for the paper. Detailed comments are below:

Major concerns:

1) Many of the results are dependent on one specific antibody and it would appear that different antibodies give slightly different results in some of these assays (both in this manuscript and the co-submitted manuscript by Ribeiro et al). Can the authors compare antibodies in both western blot and fluorescence assays to be fully transparent about when there is an IFN-gamma-induced increase in ADPr? Detection of this signal in an antibody-specific way does not discount its existence, but it would be useful for future studies for the authors to be clear about when this can be observed and when it can not and even provide some speculation about the reasons for those differences.

The reviewer has raised a concern regarding the use of different antibodies, and if they indeed show an increase in ADPr after IFNg treatment. Our western blot analysis has been completed using an newly developed antibody from the Matic laboratory (PMCID: PMC10205078) that is also since August commercially available (Biorad : AbD43647). While we have currently exclusively used during our western blotting analysis, we have included two different ADPr antibodies for immunofluorescence analysis: Fig 2A, E and 4B are using the poly/mono ADPr from Cell Signaling Technology (CST#83732) while Fig 2 C, D and 4A use the mono-ADPr specific antibody. Both clearly shown an increase in ADPr after IFNg treatment that is abolished upon PARP14 inhibition or depletion. Indeed,

we intentionally included IF data using both antibodies to illustrate that the phenotype we observed upon depletion of PARP9/DTX3L/PARP14 was not an antibody specific phenotype.

Moving forward with the revision, we will indeed perform western blotting analysis to compare IFNg induced ADP-ribosylation using several different antibodies including poly/mono ADPr antibody from CST, in addition to using ADP-ribose binding reagents from Millipore which are commonly used regularly in ADPr studies.

2) The authors do not formally show that the ADPr signal that they detect is directly catalyzed by PARP14. Given all of the cross-regulation of PARPs, and the dependence of these enzymes on each other (some of which is nicely shown in this paper), it would be more appropriate to be interpret these results as indicating that PARP14 activity is required for this increase in ADPr signal rather than that is the "main enzyme responsible for this modification".

We can appreciate this comment from the reviewers and will revise this statement. Additionally, we plan to extend our studies based on this comment and examine how depletion of other interferon induced PARPs such as PARP12 (which is mentioned in point 4 by this reviewer) regulates IFNg induced ADPr as compared to PARP14/PARP9/DTX3L.

3) The authors go into great molecular detail about a predicted AlphaFold model of a complex between PARP14, DTX3L, and PARP9. However, they do not validate this model in any way. The atomic model they describe should provide predictions for specific mutations that can be made to disrupt the interaction and thereby validate their predicted AlphaFold model (e.g. their detailed Fig. 5C and 5D). Simply removing the N-terminal half to 2/3 of DTX3L and showing that this interaction is lost is not sufficient to validate that model, and does not justify overly interpreting a predicted AlphaFold model.

We will perform mutagenesis and produce truncations of DTX3L (in particular involving KH2-3 domains) and perform functional assays as in Fig 5 to further support the model and confirm the interacting domains (as we did for PARP14 that narrowed down the interaction to KH-WWE-CAT construct).

4) In many cases, the authors are careful to not overinterpret the data on SARS-CoV-2 Mac1. However, this does not extend to the abstract. Their data do not provide "the mechanism by which Mac1 counteracts the activity of antiviral PARPs". Their data show that overexpressed Mac1 is able to reverse ADP-ribosylation, which is a result that has been shown repeatedly on a variety of ADP-ribosylated substrates. Thus, removal of the IFN-induced ADPr is one potential mechanism for Mac1 function, but by far not the only potential mechanism. Do other overexpressed macrodomains not show the same decrease in ADPr signal? Does infection with a Mac1-active, but not Mac1-inactive virus, cause this same decrease in ADPr signal? These types of experiments would get the authors closer to that kind of mechanistic interpretation. Notably, a recent paper (<https://pubmed.ncbi.nlm.nih.gov/37695054/>) showed that Mac1 mutant virus is inhibited in vivo in a PARP12-dependent manner. Based on this paper, which the authors should cite, as well as many other potential mechanisms that have been proposed for Mac1 activity, the authors should be much more cautious in their interpretation of their macrodomain data.

We indeed have tried to be careful to not overinterpret our data regarding SARS-CoV-2 Mac1 throughout the main text of our manuscript. We will also make sure that this now extends to what is said in the abstract.

In regards to the comment: do other overexpressed macrodomains not show the same decrease in ADPr, we are currently exploring this further, specifically by looking into how PARP14 and PARP9 MD1 will regulate IFN γ induced ADPr by immunofluorescence. Moreover, our supplemental figure S3 shows that overexpression of PARP9 (that harbours a hydrolytic macrodomain) is capable of removing PARP14 derived ADPr in a PARP14 overexpression system. We plan to extend our studies to investigate if this also holds true for interferon induced ADPr by endogenous PARP14.

However, we feel that extending our studies into examining responses to SARS-CoV-2 Mac1 mutant viruses is currently outside the scope of this paper. We aim to address these reviewer concerns with extended discussion within the manuscript.

5) The data on ubiquitin in Figs. 4B and EV4 are very preliminary. It is difficult to draw any meaningful conclusions from single images that contain <10 cells, only one or two are magnified sufficiently to see any foci or colocalization. Using those images alone, one is unable to conclude that the authors have presented "the first visualisation of ADPr-dependent ubiquitylation in the IFN response". Additional images and quantification are required if the authors want to conclude this.

We appreciate that these images preliminary and can be improved to support our conclusions. Our statistical analysis shows that depletion of PARP9 increases ubiquitin foci number in response to IFN γ . In addition to planning to provide improved images and quantifications as requested, we already have a complementary set of data showing the increase in ubiquitin foci colocalising with mono-ADPr upon PARP9 depletion using both different ADPr and ubiquitin antibodies (i.e. antibodies different to ones utilised in the current manuscript version). Please see the images pasted below. Moreover, we have additional data showing that co-depletion of PARP9 with PARP14 does not increase the ubiquitin foci, which suggests that PARP14 dependent-ADPr is upstream of ubiquitination. We will also try the initial characterisation of the ubiquityl and ADP-ribosyl adducts by chemical and enzymatic treatments to assess their reversibility by specific enzymes and the nature of the chemical bonds.

Figure: PARP14, PARP9 and DTX3L regulate ADPr and ubiquitin foci formation. (A) Widefield images showing A549 WT cells deleted of PARP9, DTX3L or PARP14 in the presence or absence of IFN γ . Cells were stained with Hoechst (Blue) and mono-ADPr (Green) and Ubiquitin (Red).

Other concerns:

1) The antibody used for PARP14 in Figure 2 does not appear to be entirely specific for PARP14 given how much signal remains in the PARP14 KO cells. Do the authors have another way to support the claim that PARP14 co-localizes with ADPr signal in those cells? We can appreciate that the level of background can be concerning to the reviewer, however this signal is severely reduced in PARP14 KO strongly suggesting that some PARP14 is present at these sites. We will also test additional PARP14 antibodies for their effectiveness in IF. However, whether PARP14 remains at the ADPr sites that it makes or not does not affect other conclusions of the manuscript.

Referee #3 (Report for Author)

The study from the Ahel group addresses the contribution of PARP14 to IFN-induced ADP-ribosylation in cell lines, and possible interplay with DTX3L/PARP9. There are two approaches that led to the primary observations described in the manuscript. First, a recently developed antibody probe for mono-ADP-ribosylation was used in blotting experiments to identify (by molecular weight) several polypeptides that undergo PARP14 activity-dependent ADP-ribosylation in response to IFN treatment, which has been shown on other studies to induce PARP14 expression. The same antibody was used in fluorescence microscopy experiments to show the ADP-ribose signal generated by PARP14 localizes in a perinuclear zone that in some cells is visualized as small foci. Second, biochemical data is presented that shows PARP14 activity can be regulated by PARP9 via a hydrolysis-active macrodomain in the latter. In this manner, PARP9 bound to PARP14, or DTX3L/PARP9 bound to PARP14, is envisioned to modulate PARP14 output. Because PARP14 and DTX3L/PARP9 are all induced by IFN signaling, the data suggest that PARP14 output may be titrated by an "opposing" level of DTX3L/PARP9, though other scenarios (including effects on the levels of specific substrates) can be envisioned. Overall the experiments in the manuscript are well executed but some additional data will help solidify the molecular mechanism and reveal the importance of the biochemical observations with regard to IFN signaling.

Figure 1, 2: It is a surprising that despite two figures with very nice quality blotting and IF are shown with the new antibody, but we are still left with no knowledge as to the identity of the PARP14 substrates. The authors infer that PARP14 is a major substrate based on MW, and this should be verified by IP and blotting. Given that different labs even within the field are using different reagents for detection of ADP-ribose, it would be helpful if the specific antibody name was used in the figure panels throughout the paper, or minimally in the figure legends. Also, p-STAT1 blotting is shown but there was no comment in the results on the effect of PARP14i (or lack thereof) as the data especially in Figure 1C indicates PARP14 has no apparent role in JAK-STAT signaling - at least through STAT1. Consistent with this view, Figure 3C shows that neither PARP14 KO nor DTX3L siRNA (and the associated loss of PARP9) has no obvious effect on STAT1 and p-STAT1 induction by IFN. The former also does not affect IFN induction of DTX3L/PARP9. These data would seem to rule out a PARP14 contribution to the core signaling and transcription events that follow IFN treatment.

Please comment. Also, statistical tests should be applied to the image quantitation, which is shown.

We are glad that Reviewer 3 can appreciate the latest antibody in the ADPr field and how it has allowed visualisation of protein substrates that can be ADP-ribosylated in response to immune stimuli. We recognise that there is the need for identification of the ADP-ribosylated substrates. We will perform the IP as requested to confirm the identity of PARP14 as one of the ADP-ribosylated proteins.

In addition to this, we recently published a list of ADP-ribosylated proteins that are potentially regulated by PARP14 catalytic domain and macrodomain 1 (Dukic et al, 2023). We will address by western blotting whether knockdown of some of these factors (eg. p62, RPA70, HDAC2) results in decreased mono-ADPr signal of proteins the same molecular weight as the targets.

We would like to note that we are collaborating with Michael Nielsen (the world leading expert in mass spectrometry approaches to study ADP-ribosylation) on identifying the exact sites in proteins modified by PARP14 in cells, but given that identification of the exact sites in proteins on glutamate residues is extremely challenging, this is something beyond the scope this manuscript. The data and tools developed by this manuscript will surely accelerate these efforts for the benefit of the whole field of research.

We will comment of the lack of effect of PARP14/DTX3L on STAT1 signalling in our models, however, this is not surprising given relatively small effects on published models and divergence of these signalling pathways in different cell types. Indeed, the effect of PARP14 on STAT1 phosphorylation has different effects in different studies. For example, in Iwata et al 2016 they show that depletion PARP14 increases p-STAT1 levels after immune stimulation, while Caprara et al 2018 showed the opposite effect where depletion of PARP14 decreased phospho-STAT1 after LPS treatment. Unpublished data from Mario Niepel and Ribon therapeutics show no major effect of PARP14 inhibitors on transcription in most of the models tested.

We will do statistical tests as advised and sort out the antibody nomenclature.

Figure 3: The authors should clarify if there are cell-line or technical differences in terms of DTX3L/PARP9 influence on PARP14 expression (e.g. siRNA vs KO's). Also, with the data spread (Fig. 3D) it is hard to tell if siDTX3L affects PARP14 message levels +IFN in A549 cells.

We will address the reviewers concerns and clarify them by additional repeats accordingly.

Figure 4: It may not be obvious how to reconcile the images with the treatment conditions, particularly a comparison of panel A vs panel B. PARP9 depletion appears to have minimal or no effect on IFN-induced mono-ADP-ribosylation in either panel, yet PARP9 MD modulation of PARP14 catalytic activity is the model put forth in the manuscript.

To better address the concerns of the reviewer in this figure, we aim to quantify ADPr signal to provide statistical levity to the statements we have made. Additionally, we aim to clarify our statements within the text.

Figure5: Data are presented to show the MD of PARP9 can remove ADP-ribose from PARP14. Unless I am misunderstanding the experimental design, using PARP14 auto-modification state as the readout for PARP14 activity in this case may not be sufficient to draw conclusions regarding an effect of PARP9 on PARP14 activity. Since the treatment condition (supplementing the second phase of the reaction with a MD, which hydrolyzes ADP-ribose from auto-modified PARP14) would release ADP-ribose even from fully-active PARP14, it is not obvious how this event necessarily modulates PARP14 activity - unless the specific auto-ADP-ribosylated sites in PARP14 are mapped, mutated, and shown to be necessary for catalytic activity. It seems the data that points to a role for DTX3L/PARP9 in modulating PARP14 is Figure EV3B, where siRNA was used against PARP9 and mono-ADP-ribosylation induced by Poly(I:C) followed by blotting. In U2OS cells, PARP9 KD had little or no effect on PARP14 mono-ADP-ribosylation (lanes 2, 4). In A549 cells, PARP9 KD probably had some effect on the intensity of some bands labeled by PARP14 (lanes 7, 9). The most convincing effect was on RPE1 cells, where PARP9 KD increased the intensity of the ~51 kDa band (lanes 12, 14). There is a slight concern about using the ADPr readout on blots (without knowing if the substrate protein levels are changing) since DTX3L/PARP9 depletion could have effects independent of PARP14 that are manifest as ADP-ribosylation differences but actually involve changes in protein levels via E3-mediated degradation, or loss thereof. This gets back to the point that if knowing the specific substrates labeled by PARP14 (such as the 51 kDa band) would position the group for a a more robust test of the models.

Regarding this point by the reviewer, we would like to clarify what we have tested and what we show in Figure 5 as it seems that the reviewer partially misunderstood our result: We are trying to say here that for the forward reaction it is not PARP9 that affects PARP14 (catalytic) activity, but instead, DTX3L does. Fittingly, our model suggests only direct contact of PARP14 with DTX3L (and not PARP9, see Fig 3B, EV3A) and we show that the DTX3L alone inhibits PARP14 activity (Fig 5B, E). At the second tier of regulation, PARP9 hydrolyses PARP14 ADPr – and so explains the increased MAR signal upon PARP9 KD. In our supplementary Figure EV3C we already show that PARP9 alone can dramatically reduce PARP14 ADPr in an overexpression model.

We again note the reviewers concerns regarding identification of specific targets, and would therefore refer to comments under figure 1,2 where we discuss approaches to identify/validate targets.

Lastly, to extend and validate these findings in cells, the authors should be able to reconstitute PARP9 KO cells with WT and PARP MD1 hydrolysis mutant constructs and determine the effect on PARP14 auto-modification and PARP14 activity towards the substrates shown by blotting in Figure 1. This will provide a cell-based test of the model for DTX3L/PARP9 regulation of PARP14, and it will also shed light on the relative contributions of simple binding to PARP14 versus removal of ADP-ribose from PARP14.

As mentioned above, our model does not suggest that PARP9 binds to PARP14 directly, and that binding effect is via DTX3L. We can nevertheless check by complementation whether PARP9 MD1 activity affects PARP14 activity. As we currently do not have access to A549 PARP9KO cells, we plan to combine siRNA depletion PARP9 with expression of siRNA resistant PARP9 WT or MD1 mutant. Similarly, we will use this approach to test changes to ADPr-dependent ubiquitination. We will also confirm PARP9 effect on PARP14 activity and

the importance of the MD1 activity by our PARP14 overexpression system in PARP14-deficient 293T cells (Dukic et al, 2023).

Minor comment on Figure EV1: PARP14 protein stability cannot be assessed from a single timepoint.

To address this point by the reviewer we will perform a timecourse to better assess changes in protein stability.

References

Caprara, G., Prosperini, E., Piccolo, V., Sigismondo, G., Melacarne, A., Cuomo, A., Boothby, M., Rescigno, M., Bonaldi, T., and Natoli, G. (2018). PARP14 Controls the Nuclear Accumulation of a Subset of Type I IFN-Inducible Proteins. *J Immunol* 200, 2439-2454.

Đukić, N., Strømmland, Ø., Elsborg, J.D., Munnur, D., Zhu, K., Schuller, M., Chatrin, C., Kar, P., Duma, L., Suyari, O., et al. (2023). PARP14 is a PARP with both ADP-ribosyl transferase and hydrolase activities. *Science Advances* 9, eadi2687.

Iwata, H., Goettsch, C., Sharma, A., Ricchiuto, P., Goh, W.W.B., Halu, A., Yamada, I., Yoshida, H., Hara, T., Wei, M., et al. (2016). PARP9 and PARP14 cross-regulate macrophage activation via STAT1 ADP-ribosylation. *Nature Communications* 7, 12849.

Dr. Ivan Ahel
University of Oxford
Sir William Dunn School of Pathology
South Parks Road
Oxford OX1 3RE
United Kingdom

15th Dec 2023

Re: EMBOJ-2023-115820
PARP14 and PARP9/DTX3L regulate interferon-induced ADP-ribosylation

Dear Ivan,

Thank you for sending me your detailed tentative responses and revision plan for your recent EMBO Journal submission on PARP14 and IFN-induced ADP-ribosylation. I have now had a chance to carefully consider them, and appreciate that your answers appear clarify the majority of the raised concerns, including key points of the referees. We shall therefore be happy to formally invite a new version of the manuscript, revised and extended along the lines proposed in your response letter. In particular, the additional experiments to be included in response to ref 2 points 1, 2, 3, 5, as well as in response to ref 3, would appear very valuable for the revision. I also understand that studies with Mac1-mutant viruses (ref 2 pt 4) or identifying exact modification sites in target proteins (ref 3) would lie outside the scope of the present submission.

Should you need an extended revision period, please let me know, and as always, competing manuscript published during the course of this revision will not affect our final decision on your study. But please be reminded that in light of our single-major-revision-round policy, it will be important to convince the critical referees with the additional data and clarifications at this stage.

Please also note the additional information and more detailed guidelines on how to prepare a revision below (and in our online Guide to Authors) - closely adhering to them shall greatly facilitate the editorial process at the time of resubmission.

Thank you again for the opportunity to consider this work, and I look forward to receiving your revised manuscript in due time.

With kind regards,

Hartmut

9) Digital image enhancement is acceptable practice, as long as it accurately represents the original data and conforms to community standards. If a figure has been subjected to significant electronic manipulation, this must be clearly noted in the figure legend and/or the 'Materials and Methods' section. The editors reserve the right to request original versions of figures and the original images that were used to assemble the figure. Finally, we generally encourage uploading of numerical as well as gel/blot image source data; for details see: embopress.org/page/journal/14602075/authorguide#sourcedata

At EMBO Press, we ask authors to provide source data for the main manuscript figures. Our source data coordinator will contact you to discuss which figure panels we would need source data for and will also provide you with helpful tips on how to upload and organize the files.

In the interest of ensuring the conceptual advance provided by the work, we recommend submitting a revision within 3 months (14th Mar 2024). Please discuss the revision progress ahead of this time with the editor if you require more time to complete the revisions. Use the link below to submit your revision:

Link Not Available

We would like to extend our thanks to the reviewers and the editor for taking the time and providing comments to help improve our manuscript. We have provided here a point-by-point response to the reviews, with our comments shown in blue.

Referee #1 (Report for Author)

The manuscript by Kar et al. establishes PARP14 as the primary member of the PARP family accountable for IFN γ (and other immune activators)-stimulated cellular ADP-ribosylation. The researchers also present supporting evidence suggesting that PARP9/DTX3L regulates PARP14 levels in cells by directly regulating PARP14's catalytic activity. They use alphaFold modeling to suggest how PARP9/DTX3L could bind to PARP14 and regulate its catalytic activity. They then show in vitro that PARP9/DTX3L can inhibit the catalytic activity of PARP14. This manuscript is well-crafted and provides keen insight into IFN γ -stimulated mono-ADP-ribosylation and the regulation of PARP14 catalytic activity. The increasing interest in PARP14 as a therapeutic target makes this study timely and should be of broad interest to the EMBO readership. Some minor points should be addressed prior to publication:

We thank the reviewer for their comments regarding our manuscript and appreciate their suggestions to improve our manuscript further.

1. Fig. 1A: the PARP14i + IFN γ WT condition seems missing.

We hope that the reviewer can appreciate that these conditions were included in both Figure 1B and 1C. However, we also now include additional data showing ADP-ribosylation in WT A549 cells +/- PARP14i in the presence or absence of IFN γ in Figure EV1A. Moreover, we compare the ADP-ribosylation signal in these conditions using the mono-specific ADPr antibody in addition to the poly/mono ADPr antibody from Cell Signaling technologies (CST #83732) which is a commonly used antibody to detect ADP-ribosylation.

2. Fig. 1 title: states that PARP14 is the major ART activated in response to immune stimuli. This could be rephrased to say that PARP14 catalytic activity is required for ADP-ribosylation induced by IFN γ and other immune activators. It is hard to say if these stimuli "activate" PARP14 as they also increase PARP14 protein levels.

We have changed the title of the figure legend to "Immunity responses induce PARP14-dependent ADP-ribosylation".

3. Fig EV1: the ADPr Western blot should be shown.

We have extended this figure by providing additional timepoints and included ADPr blot with experiment (Fig EV1F).

4. Fig. 2A: Does the ADPr signal they observe co-localize with mitochondria? The authors could do a co-staining experiment with mitochondrial markers to address this.

We thank the reviewer for this suggestion. We agree that the signal could resemble mitochondrial networks, however, this does not appear to be the location of the ADPr signal. We performed immunofluorescence for ADPr and compared it to mitochondria stained with mitotracker and it is clear that there is no colocalization between the two signals.

5. Fig. 4B: the authors should quantify the co-localization of ubiquitin with ADPr.

We thank the reviewer for this suggestion. In order to provide quantification, we repeated the experiment and have taken higher resolution images using confocal microscopy. Our previous images were taken on a widefield microscope and a low magnification which made quantification difficult due to the small pixel size of the foci. The higher magnification and higher resolution images have solved this, and we have now also included colocalization data quantifying the number of ubiquitin foci that colocalise with ADPr. The quantification supports our observations that there is an increase in foci containing both modifications after PARP9 depletion and is dramatically reduced upon depletion of DTX3L or PARP14.

6. Fig. 4: can the authors speculate why the ADPr pattern changes so dramatically between the control and siPARP9 conditions (appears to change from a more clustered, perinuclear localization to a more punctate localization).

We are currently working through different hypotheses to address this comment. However, we can speculate that because PARP9 MD1 is able to hydrolyse PARP14 ADPr (Dukic et al, Sci Adv 2023), the loss of PARP9 (and its hydrolytic MD1) might preserve PARP14 ADPr, which may lead to the different localisation pattern observed especially as it could also affect the proposed ADPr dependent ubiquitination. Fittingly, we have previously shown different patterns of YFP-PARP14 WT and MD1 mutant (Đukić et al., 2023). Moreover, as ADPr has been suggested to be a catalyst of phase separation (Rhine et al., 2022), one could speculate that the increased ADPr seen on western blots with the PARP14 MD1 mutant could promote the accumulation of ADP-ribose or ADP-ribosylated proteins into these more punctate foci. If this is the case, it could suggest that these structures may not stain equally well with antibody during IF and could thus account for the discrepancies between our immunofluorescence data and our western blots. We could also speculate that if PARP14/DTX3L/PARP9 complex bind to other partners, and that these interactions could be perturbed upon PARP9 loss and thus we have a change in structure when the large complex is disrupted. We have currently not discussed these possibilities within the manuscript as they are too speculative, and we believe the reasoning requires proper experimental investigation and currently lies outside the scope of the current manuscript.

Referee #2 (Report for Author)

Kar et al. show that addition of IFN-gamma to some cells (A549 and RPE1, but not U2OS) results in increased cellular ADP-ribosylation (ADPr) as detected by a new anti-ADPr antibody. They show that this increased ADPr signal is dependent on the catalytic activity of PARP14 and can be reduced by overexpression of a macrodomain (Mac1) from SARS-CoV-2. They further show that PARP14 activity is regulated by auto-ADP-ribosylation as well as interactions with the PARP9/DTX3L complex. Finally, they provide microscopy data that cytoplasmic ubiquitin co-localizes and is dependent on IFN-induced cytoplasmic ADPr.

There are several interesting aspects to this manuscript. The authors provide clear and convincing data on the requirement for PARP14 in the IFN-induced increase in ADPr signal. In addition, they clearly show that PARP14 activity is regulated by both PARP14 ADP-ribosylation as well as the presence of PARP9/DTX3L. Both of these pieces of data further support the model that has been emerging from their work and others that PARP14, PARP9, and DTX3L are important regulators of the IFN-gamma response. However, in several places, the authors overinterpret their data, drawing conclusions that are not supported by the data, which weakens enthusiasm for the paper. Detailed comments are below:

We thank the reviewer for their careful review of our manuscript. We have taken care to address the points put forward by the reviewer.

Major concerns:

1) Many of the results are dependent on one specific antibody and it would appear that different antibodies give slightly different results in some of these assays (both in this manuscript and the co-submitted manuscript by Ribeiro et al). Can the authors compare antibodies in both western blot and fluorescence assays to be fully transparent about when there is an IFN-gamma-induced increase in ADPr? Detection of this signal in an antibody-specific way does not discount its existence, but it would be useful for future studies for the authors to be clear about when this can be observed and when it can not and even provide some speculation about the reasons for those differences.

We thank the reviewer for this comment. In our original submission we had already included data using the mono-ADPr specific antibody from the Matic lab (now commercially available from Biorad) and a second poly/mono antibody from Cell Signaling Technologies (CST #83732), at least in IF studies. Each antibody was able to detect the cytoplasmic ADPr after interferon stimulation and a similar effect on ADPr was seen with depletion of DTX3L, PARP9 and PARP14 with each antibody (Fig 4 A, B), demonstrating that this signal is not antibody specific.

We have also now included western blot data in A549 (Fig EV1A) and U2OS DTX3L KO cells (Fig EV3B), comparing the mono-ADPr antibody with the poly/mono antibody from CST (#83732). While we were able to detect an increase in ADPr signal after interferon stimulation in A549 cells using the poly/mono antibody, the increase was better observed using the mono-ADPr antibody (Fig EV1A). Furthermore, the poly/mono antibody performs relatively poorer in the cell lines that have higher level of endogenous PARP1-dependent long poly(ADP-ribose) chains like U2OS (Fig EV3B). In these cells, we were not able to convincingly detect the increase in ADPr after IFN γ stimulation using the poly/mono ADPr

antibody (likely due to a strong background from PAR detection) that was clearly detected using the mono-ADPr antibody. This highlights the importance of this newly described antibody which will no doubt continue to expand our current knowledge of monoADPr. To increase transparency, we have now indicated on both the western blots, and in figure legends, the catalogue number of antibody used in each panel.

2) The authors do not formally show that the ADPr signal that they detect is directly catalyzed by PARP14. Given all of the cross-regulation of PARPs, and the dependence of these enzymes on each other (some of which is nicely shown in this paper), it would be more appropriate to be interpret these results as indicating that PARP14 activity is required for this increase in ADPr signal rather than that is the "main enzyme responsible for this modification".

We thank the reviewer for this concern. We have also now provided additional data that another interferon stimulated PARP, PARP12, does not affect IFN γ stimulated ADPr (Fig EV1C-D). Nevertheless, we can appreciate the need to tone down our statement here. Thus, we have changed this sentence to now read "Together, these results show that PARP14 ART activity is activated in response to a wide range of immune stimuli, and that its ART activity is a major contributor in the production of mono-ADPr in these conditions".

3) The authors go into great molecular detail about a predicted AlphaFold model of a complex between PARP14, DTX3L, and PARP9. However, they do not validate this model in any way. The atomic model they describe should provide predictions for specific mutations that can be made to disrupt the interaction and thereby validate their predicted AlphaFold model (e.g. their detailed Fig. 5C and 5D). Simply removing the N-terminal half to 2/3 of DTX3L and showing that this interaction is lost is not sufficient to validate that model, and does not justify overly interpreting a predicted AlphaFold model.

We thank the reviewer for their constructive feedback. The AlphaFold2 model predicts that the interaction is between DTX3L KH2-3 and PARP14 KH8-WWE-ART domains. We have now generated specific mutations along this interface: on PARP14 KH8 (D1466R, D1468R, E1469R, K1491E, E1525R, D1529R, E1533R) and on DTX3L KH2 (E292R, Q299R), as shown in Fig 5F (also copied below). Mutations on DTX3L FL^{E292R, Q299R} only partially inhibits PARP14 FL and PARP14 KH8-WWE-ART automodification activity. To complement this, the automodification of PARP14 KH8-WWE-ART bearing multiple mutations on its KH8 domain (PARP14 KH8^{mut}-WWE-ART) are not affected by DTX3L FL and DTX3L FL^{E292R, Q299R} (Fig 5G, also copied below). These results suggest that the inhibition of PARP14 activity relies on the interaction interface between DTX3L KH2 and PARP14 KH8 as predicted by the AlphaFold2 model. We noted that the double mutations on DTX3L FL^{E292R, Q299R} do not fully restore PARP14 automodification activity, which suggests that there are likely other residues/domains on DTX3L that are involved in interacting with PARP14.

Figure 5 F-G.

(F) Mutations on the interaction interface between PARP14 KH8 and DTX3L KH2 domain. (G) Auto-ADP-ribosylation reaction of PARP14 FL, PARP14 KH8-WWE-ART, and PARP14 KH8^{mut}-WWE-ART performed with ³²P-NAD⁺ in the presence of DTX3L FL or DTX3L FL mutant. Each experiment has been completed in triplicate.

4) In many cases, the authors are careful to not overinterpret the data on SARS-CoV-2 Mac1. However, this does not extend to the abstract. Their data do not provide "the mechanism by which Mac1 counteracts the activity of antiviral PARPs". Their data show that overexpressed Mac1 is able to reverse ADP-ribosylation, which is a result that has been shown repeatedly on a variety of ADP-ribosylated substrates. Thus, removal of the IFN-induced ADPr is one potential mechanism for Mac1 function, but by far not the only potential mechanism. Do other overexpressed macrodomains not show the same decrease in ADPr signal? Does infection with a Mac1-active, but not Mac1-inactive virus, cause this same decrease in ADPr signal? These types of experiments would get the authors closer to that kind of mechanistic interpretation. Notably, a recent paper (<https://pubmed.ncbi.nlm.nih.gov/37695054/>) showed that Mac1 mutant virus is inhibited in vivo in a PARP12-dependent manner. Based on this paper, which the authors should cite, as well as many other potential mechanisms that have been proposed for Mac1 activity, the authors should be much more cautious in their interpretation of their macrodomain data.

We appreciate what the reviewer has pointed out. We have completed our studies using expressed SARS2-Mac1, and not in a viral context. To ensure we are not overinterpreting the

SARS2 Mac1 data, we have toned down our statement in the abstract, now suggesting that this is a possible mechanism that could regulate PARP14 dependent ADPr under such condition. We apologise for the oversight of the PARP12 reference and have now included it in our manuscript. Furthermore, we also examined if PARP12 could contribute to interferon induced protein ADPr to a comparable level as PARP14 depletion but did not observe a decrease in ADPr upon PARP12 depletion (new figure EV1C,D)

We have now mentioned Mac1 hydrolase-dependent and Mac1 hydrolase-independent mechanisms in the discussion of the manuscript.

5) The data on ubiquitin in Figs. 4B and EV4 are very preliminary. It is difficult to draw any meaningful conclusions from single images that contain <10 cells, only one or two are magnified sufficiently to see any foci or colocalization. Using those images alone, one is unable to conclude that the authors have presented "the first visualisation of ADPr-dependent ubiquitylation in the IFN response". Additional images and quantification are required if the authors want to conclude this.

We have taken care to improve the quality of our ubiquitin/ADPr data. In order to provide quantification, we repeated the experiment and have taken higher resolution images using confocal microscopy to improve resolution of the foci. Our previous images were taken on a widefield microscope and a low magnification which made quantification difficult due to the small pixel size of the foci. The higher magnification and higher resolution images have solved this and we have now also included colocalization data quantifying the number of ubiquitin foci that colocalise with ADPr. The quantification supports our observations that there is an increase in foci containing both modifications after PARP9 depletion and is dramatically reduced upon depletion of DTX3L or PARP14.

We believe that the images that are now included as Figure 4B and Figure EV4A allow for easier visualisation of the foci and colocalization between ubiquitin and ADPr. Furthermore, we have quantified the number of ubiquitin foci in each condition that show colocalization with ADPr. We are able to see a significant increase in ubiquitin/ADPr staining foci in siPARP9 conditions. Furthermore, we have repeated the immunofluorescence experiment using the mono-ADPr antibody (instead of poly/mono ADPr antibody from CST which was used in Fig 4B, Fig EV4A) and a different ubiquitin antibody from another company and which was raised in a different host species (Fig EV4B). We were again able to recapitulate the experimental results, seeing an increase in foci staining for both ubiquitin and mono-ADPr upon depletion of PARP9.

Other concerns:

1) The antibody used for PARP14 in Figure 2 does not appear to be entirely specific for PARP14 given how much signal remains in the PARP14 KO cells. Do the authors have another way to support the claim that PARP14 co-localizes with ADPr signal in those cells?

We repeated the experiment using confocal microscopy instead of widefield microscopy which was used in the original submission. The new staining and imaging we were able to decrease the amount of background signal from the PARP14 antibody. Furthermore, we quantified the amount of PARP14 signal in WT cells in the presence or absence of interferon stimulation and in the presence of PARP14i. We were able to see the same trend we see on western blotting that PARP14 levels increase after IFN γ treatment, and that the total cellular PARP14 levels increased after PARP14i treatment. Furthermore, we quantified the PARP14

signal in PARP14KO cells with and without IFN γ treatment. Here we saw no change in the level of PARP14, and that the overall signal from the PARP14 antibody in the KO cells was less than the basal levels of PARP14 in unstimulated WT cells. Finally, we performed the immunofluorescence with a second PARP14 antibody and were able to see again the same trend of PARP14 expression and cellular localisation after interferon stimulation, and that the PARP14 signal colocalised with mono-ADPr after stimulation (Fig EV2B,C)

Referee #3 (Report for Author)

The study from the Ahel group addresses the contribution of PARP14 to IFN-induced ADP-ribosylation in cell lines, and possible interplay with DTX3L/PARP9. There are two approaches that led to the primary observations described in the manuscript. First, a recently developed antibody probe for mono-ADP-ribosylation was used in blotting experiments to identify (by molecular weight) several polypeptides that undergo PARP14 activity-dependent ADP-ribosylation in response to IFN treatment, which has been shown on other studies to induce PARP14 expression. The same antibody was used in fluorescence microscopy experiments to show the ADP-ribose signal generated by PARP14 localizes in a perinuclear zone that in some cells is visualized as small foci. Second, biochemical data is presented that shows PARP14 activity can be regulated by PARP9 via a hydrolysis-active macrodomain in the latter. In this manner, PARP9 bound to PARP14, or DTX3L/PARP9 bound to PARP14, is envisioned to modulate PARP14 output. Because PARP14 and DTX3L/PARP9 are all induced by IFN signaling, the data suggest that PARP14 output may be titrated by an "opposing" level of DTX3L/PARP9, though other scenarios (including effects on the levels of specific substrates) can be envisioned. Overall the experiments in the manuscript are well executed but some additional data will help solidify the molecular mechanism and reveal the importance of the biochemical observations with regard to IFN signaling.

We thank the reviewer for taking time to provide critical comments on our manuscript.

Figure 1, 2: It is surprising that despite two figures with very nice quality blotting and IF are shown with the new antibody, but we are still left with no knowledge as to the identity of the PARP14 substrates. The authors infer that PARP14 is a major substrate based on MW, and this should be verified by IP and blotting. Given that different labs even within the field are using different reagents for detection of ADP-ribose, it would be helpful if the specific antibody name was used in the figure panels throughout the paper, or minimally in the figure legends. Also, p-STAT1 blotting is shown but there was no comment in the results on the effect of PARP14i (or lack thereof) as the data especially in Figure 1C indicates PARP14 has no apparent role in JAK-STAT signaling - at least through STAT1. Consistent with this view, Figure 3C shows that neither PARP14 KO nor DTX3L siRNA (and the associated loss of PARP9) has no obvious effect on STAT1 and p-STAT1 induction by IFN. The former also does not affect IFN induction of DTX3L/PARP9. These data would seem to rule out a PARP14 contribution to the core signaling and transcription events that follow IFN treatment. Please comment. Also, statistical tests should be applied to the image quantitation, which is shown.

We thank the reviewer for their comments provided in order to improve our manuscript. We have taken their advice and have completed several additional experiments and implemented several changes throughout to support our initial submission. Firstly, we have performed an immunoprecipitation experiment to add validity to our inference that PARP14 is a major target of ADPr after interferon stimulation (Fig EV1B). Here we immunoprecipitated

endogenous PARP14 from A549 cells in the presence or absence of PARP14i, with and without interferon stimulation. As expected, we see an increase in PARP14 levels after IFN γ treatment, which is further increased upon treatment with PARP14i. Our data shows that immunoprecipitated PARP14 is ADP-ribosylated after IFN γ , and that this is lost with PARP14i treatment, and 200 kDa band likely represents PARP14.

Secondly, we have added specific antibody names or catalogue numbers of the different ADPr antibodies both on western blot panels and throughout the figure legends. Furthermore, we have now included additional data (Fig EV1A, Fig EV3B) directly comparing the two ADPr antibodies used throughout this study (Mono-ADPr (from the Matic lab, now available at Biorad) and poly/mono ADPr (Cell Signaling Technologies, CST #83732)) in two different cell lines. These additional panels demonstrate the advantage of the mono-ADPr antibody in western blot, being able to identify a number of targets in U2OS DTX3L KO cells that is not detectable with the poly/mono ADPr CST antibody. The poly/mono antibody appears to perform relatively poorer in the cell lines that have higher level of endogenous PARP1-dependent long poly(ADP-ribose) chains like U2OS (Fig EV3B). In these cells, we were not able to convincingly detect the increase in ADPr after IFN γ stimulation using the poly/mono ADPr antibody (likely due to a strong background from PAR detection) that was clearly detected using the mono-ADPr antibody.

Within the literature, there is no unifying account on the current role of PARP14 regulating changes in phospho-STAT1 levels, with the effect of PARP14 on STAT1 phosphorylation showing no real consistencies across different studies. For example, in Iwata et al 2016 they show that depletion of PARP14 increases p-STAT1 levels after immune stimulation in macrophages and THP-1 cells, while Caprara et al 2018 showed the opposite effect where depletion of PARP14 decreased phospho-STAT1 after LPS treatment in murine macrophages. Moreover, the differences in these levels presented often show very small changes between WT and PARP14-deficient or depleted cells. In our hands we do not observe any significant, reliable effect on p-STAT1 levels in PARP14 or DTX3L depleted cells, which may be explained by the use of different cell lines as compared to already published literature. Given the inconsistencies within the literature, we have not made further comments on this within the manuscript.

Finally, we have made improvements on some of our imaging in Figure 2, moving from a widefield microscope to a confocal microscope in order to improve resolution of our images, and reduce the background given from some antibodies. Furthermore, we have quantified PARP14, PARP9 and mono-ADPr levels before and after interferon stimulation and provided statistical analysis for the new and previously included experiments.

Figure 3: The authors should clarify if there are cell-line or technical differences in terms of DTX3L/PARP9 influence on PARP14 expression (e.g. siRNA vs KO's). Also, with the data spread (Fig. 3D) it is hard to tell if siDTX3L affects PARP14 message levels +IFN in A549 cells.

Rebuttal Figure 2: qPCR analysis of PARP14 gene expression. (A) Mean PARP14 gene expression levels from three independent biological replicates. (B) The three individual biological replicates obtained from quadruple technical replicates.

We also appreciate that the qPCR data we have included in Fig 3D, although not statistically significant, could raise questions. Therefore, we have repeated the experiments with a further three biological replicates and quadruple technical replicates (Rebuttal Figure 2B). In each iteration we see no difference in PARP14 expression after interferon stimulation upon depletion of DTX3L. We now include the mean PARP14 gene expression data seen in the rebuttal figure 2A as the new Fig 3D.

When we examine our blots, we see a small decrease in PARP14 protein levels with depletion of DTX3L in A549 (which we have commented on in our original manuscript). This appears to be similar in RPE1 cells, while there seems to be no significant differences in PARP14 levels in U2OS cells upon depletion or KO of DTX3L. We have not commented upon this, however, as the differences are small and could come down to technical reasons or cell line specific differences.

Figure 4: It may not be obvious how to reconcile the images with the treatment conditions, particularly a comparison of panel A vs panel B. PARP9 depletion appears to have minimal or no effect on IFN-induced mono-ADP-ribosylation in either panel, yet PARP9 MD modulation of PARP14 catalytic activity is the model put forth in the manuscript.

We thank the reviewer for this comment. We based the statement primarily on western blot data which is, in this case, more quantitative and very reproducibly shows that depletion of PARP9 results in an increase of ADPr across most of the ADPr bands in the western blot compared to siCTRL (please see Fig 3G, Fig EV3C). This data also fits with new experiment included as Figure EV3D which shows that expression of PARP9 WT, but not the macrodomain 1 mutant, is capable of removing mono-ADPr. This data showing robust PARP9 MD1 activity in cells and fits our published in vitro data with recombinant proteins (Đukić et al., 2023). In figure 4, we see quite striking change in ADPr pattern upon PARP9 depletion compared to siCTRL cells. The large changes in cytoplasmic ADPr structure pose a difficulty when trying to accurately quantify the total amount of ADPr and are therefore perhaps more accurately portrayed by western blotting. Instead, we chose to describe the changes in structure. We observed that siCTRL cells show large, elongated ADPr structures after interferon stimulation that extend throughout the cytoplasm whereas PARP9 depleted cells lose the elongated structures, instead displaying small, but bright, foci throughout the cytoplasm. The change in ADPr structure observed upon PARP9 depletion could be in part due to the PARP9 MD1 activity.

Figure 5: Data are presented to show the MD of PARP9 can remove ADP-ribose from PARP14. Unless I am misunderstanding the experimental design, using PARP14 auto-modification state as the readout for PARP14 activity in this case may not be sufficient to draw conclusions regarding an effect of PARP9 on PARP14 activity. Since the treatment condition (supplementing the second phase of the reaction with a MD, which hydrolyzes ADP-ribose from auto-modified PARP14) would release ADP-ribose even from fully-active PARP14, it is not obvious how this event necessarily modulates PARP14 activity - unless the specific auto-ADP-ribosylated sites in PARP14 are mapped, mutated, and shown to be necessary for catalytic activity. It seems the data that points to a role for DTX3L/PARP9 in modulating PARP14 is Figure EV3B, where siRNA was used against PARP9 and mono-ADP-ribosylation induced by Poly(I:C) followed by blotting. In U2OS cells, PARP9 KD had little or no effect on PARP14 mono-ADP-ribosylation (lanes 2, 4). In A549 cells, PARP9 KD probably had some effect on the intensity of some bands labeled by PARP14 (lanes 7, 9). The most convincing effect was on RPE1 cells, where PARP9 KD increased the intensity of the ~51 kDa band (lanes 12, 14). There is a slight concern about using the ADPr readout on blots (without knowing if the substrate protein levels are changing) since DTX3L/PARP9 depletion could have effects independent of PARP14 that are manifest as ADP-ribosylation differences but actually involve changes in protein levels via E3-mediated degradation, or loss thereof. This gets back to the point that if knowing the specific substrates labeled by PARP14 (such as the 51 kDa band) would position the group for a more robust test of the models.

We would like to clarify what our data here shows, as we believe that there has been some misunderstanding. We aimed to determine how DTX3L could regulate the function of PARP14 given that our model from Figure 3B suggests that there is a direct interaction between PARP14 and DTX3L (this has been additionally supported by new data using specific interaction mutants in Figure 5F, G), not PARP9 and PARP14. First we confirmed in Figure 5A that PARP9 MD1 and PARP14 MD1 can efficiently remove the PARP14-dependent ADPr modification (as we have previously shown in Dukic et al 2023 Science Advances) while we also showed that DTX3L alone is unable to hydrolyse PARP14 automodification.

Next, we showed that DTX3L strongly inhibits PARP14 ADPr activity in a hydrolyse-independent manner by protein-protein interaction and this is the additional (and novel) way how PARP9/DTX3L regulates PARP14. We have now worked out the domains involved in this contact inhibition as explained below. We previously incubated PARP14 together with PARP9/DTX3L or DTX3L alone prior to adding NAD⁺, and saw an inhibition of PARP14 ART activity in the presence of DTX3L, alone or in complex with PARP9. Our data in Figure 5E indicated that this inhibition was dependent on the predicted interaction between DTX3L and C-terminal part of PARP14 (KH8-WWE-ART domains). We have now also generated specific mutations alongside the predicted DTX3L-PARP14 interface and showed that perturbing the interface between KH2-KH3 domains of DTX3L and KH8-domain-WWE-ART domains of PARP14 relieves the inhibition of PARP14 automodification activity (Figure 5F-G). The figures are also shown below. We apologise again for not emphasising the most novel aspect of our data in our previous version of the Figure 5.

Figure 5 E-G:

(E) PARP14 FL and PARP14 KH8-WWE-ART auto-ADP-ribosylation reaction performed with $^{32}\text{P-NAD}^+$ in the presence of DTX3L FL or DTX3L RD.

(F) Mutations on the interaction interface between PARP14 KH8 and DTX3L KH2 domain.

(G) Auto-ADP-ribosylation reaction of PARP14 FL, PARP14 KH8-WWE-ART, and PARP14 KH8^{mut}-WWE-ART performed with $^{32}\text{P-NAD}^+$ in the presence of DTX3L FL or DTX3L FL mutant. Each experiment has been completed in triplicate.

Altogether, we would like to emphasise that the regulation of PARP14 ADP-ribosylation is a complex system that is regulated at different levels. Indeed, our cellular data with PARP9 depletion in Figure 3 highlights the importance of the hydrolytic macrodomain in regulating cellular levels of PARP14 dependent ADPr, while our combined in vitro and cellular data suggests that the interaction of PARP14 and DTX3L can also regulate cellular ADPr through interaction-dependent inhibition of PARP14 ART activity. Moreover, we cannot rule out the possibility of additional currently unidentified proteins or substrate specific effects that could also regulate PARP14 dependent ADPr, including by ubiquitination activity of DTX3L, and we will strive to elucidate details of these complex regulations in the future. In this respect we appreciate the reviewer's comments regarding the identification of ADPr substrates. We agree that, in an ideal situation, we would know the identity of these proteins and could make more specific comments, however, we feel that this currently is outside the scope of the current manuscript, especially as identifying ADPr proteins/sites by mass spectrometry (in particular on the glutamate residues as well as looking at dual ADPr/ubiquitin modifications) are still technically very challenging.

Lastly, to extend and validate these findings in cells, the authors should be able to reconstitute PARP9 KO cells with WT and PARP MD1 hydrolysis mutant constructs and determine the effect on PARP14 auto-modification and PARP14 activity towards the substrates shown by blotting in Figure 1. This will provide a cell-based test of the model for DTX3L/PARP9 regulation of PARP14, and it will also shed light on the relative contributions of simple binding to PARP14 versus removal of ADP-ribose from PARP14.

We appreciate this comment from the reviewer, however we have been unable to create complemented PARP9 knockout cell lines within the revision time period as this would involve the preparing knock-ins and obtaining the interferon inducible PARP9 and mutants of comparable expression levels and regulation. However, we have attempted to partially satisfy these requests using a different, rather simplified system. In Figure EV3D we examine how PARP9 WT or PARP9 with a mutated hydrolytic macrodomain (PARP9 MD1 mutant) can remove PARP14-dependent ADPr in 293T cells. These cells are a natural PARP14 knockout cell line (this is the only PARP14 deficient cell line we have identified so far – as we have previously shown in Dukic et al 2023 *Science Advances*), with no detectable endogenous PARP14 detected. Furthermore, there we are unable to detect any endogenous PARP9, meaning these cells are also effectively PARP9 deficient. In Figure EV3D we express YFP-PARP14 mutated in its first macrodomain to induce large amounts of PARP14-dependent ADPr. On top of this, we have co-expressed either YFP-PARP9 WT or YFP-PARP9 MD1 mutant, making it a hydrolytic null mutant. Here we are able to see that upon PARP9 WT expression, the PARP14 dependent ADPr is reduced to almost background levels, whereas the PARP9 MD1 mutant does not remove PARP14-dependent ADPr. The equal expression of PARP9 between these two samples indicates that PARP9 cannot inhibit PARP14 ART activity by protein-protein interaction (although, as we describe above, it is not PARP9, but rather DTX3L that has an inhibitory effect on PARP14 ART activity), and thus the reduction in ADPr comes through the hydrolase activity mediated by its MD1. We wouldn't like to overinterpret these data, but at the very minimum the data show that the overexpressed PARP9 can exhibit robust hydrolase activity in cells that is dependent on its MD1 domain.

Minor comment on Figure EV1: PARP14 protein stability cannot be assessed from a single timepoint.

We have now completed an extended timecourse to look at PARP14 protein stability. This is now included as Figure EV1F.

Reference

Caprara, G., Prosperini, E., Piccolo, V., Sigismondo, G., Melacarne, A., Cuomo, A., Boothby, M., Rescigno, M., Bonaldi, T., and Natoli, G. (2018). PARP14 Controls the Nuclear Accumulation of a Subset of Type I IFN-Inducible Proteins. *J Immunol* 200, 2439-2454.

Iwata, H., Goettsch, C., Sharma, A., Ricchiuto, P., Goh, W.W.B., Halu, A., Yamada, I., Yoshida, H., Hara, T., Wei, M., et al. (2016). PARP9 and PARP14 cross-regulate macrophage activation via STAT1 ADP-ribosylation. *Nature Communications* 7, 12849.

Đukić, N., Strømmland, Ø., Elsborg, J.D., Munnur, D., Zhu, K., Schuller, M., Chatrin, C., Kar, P., Duma, L., Suyari, O., et al. (2023). PARP14 is a PARP with both ADP-ribosyl transferase and hydrolase activities. *Science Advances* 9, eadi2687.

Rhine, K., Dasovich, M., Yoniles, J., Badiee, M., Skanchy, S., Ganser, L.R., Ge, Y., Fare, C.M., Shorter, J., Leung, A.K.L., *et al.* (2022). Poly(ADP-ribose) drives condensation of FUS via a transient interaction. *Mol Cell* 82, 969-985.e911.

Dr. Ivan Ahel
University of Oxford
Sir William Dunn School of Pathology
South Parks Road
Oxford OX1 3RE
United Kingdom

30th Apr 2024

Re: EMBOJ-2023-115820R
PARP14 and PARP9/DTX3L regulate interferon-induced ADP-ribosylation

Dear Ivan,

Thank you for submitting your revised manuscript to The EMBO Journal. Two of the original referees have now assessed it once again (see comments below), and both of them are overall satisfied with your responses and revisions. We shall therefore be happy to accept the study for publication, after a final round of minor revision to incorporate the remaining presentational/statistics comments of referee 2, and to address the following editorial issues:

- Please move the funding information, currently listed in a separate section, into the article's Acknowledgement section
- Please rename the Conflict of Interest section into "Disclosure and Competing Interests Statement", in accordance with our updated Guide to Authors (<https://www.embopress.org/competing-interests>)
- As we are switching from a free-text author contribution statement towards a more formal statement based on Contributor Role Taxonomy (CRediT) terms, please remove the present Author Contribution section and instead specify each author's contribution(s) directly in the Author Information page of our submission system during upload of the final manuscript. See <https://casrai.org/credit/> for more information.
- Please adjust the format for citation of preprints as specified in our author guidelines:
The citation in the text should be: "(preprint: NAME1 et al, YEAR)"
The citation in the reference list: "Author NAME1, Author NAME2, ... (YEAR) article title. bioRxiv doi: XXX"

I am therefore returning the manuscript to you for a final round of minor revision, to allow you to make these adjustments and upload all modified files. Once we will have received them, we should be ready to swiftly proceed with formal acceptance and production of the manuscript. I would appreciate if you could send the final version back within one week from now.

Kind regards,

Hartmut

- size of the scale bars that are mandatory for all micrograph panels
- the statistical test used to generate error bars and P-values
- the type error bars (e.g., S.E.M., S.D.)
- the number (n) and nature (biological or technical replicate) of independent experiments underlying each data point

- Figures may not include error bars for experiments with $n < 3$; scatter plots showing individual data points should be used instead.

9) Digital image enhancement is acceptable practice, as long as it accurately represents the original data and conforms to community standards. If a figure has been subjected to significant electronic manipulation, this must be clearly noted in the figure legend and/or the 'Materials and Methods' section. The editors reserve the right to request original versions of figures and the original images that were used to assemble the figure. Finally, we generally encourage uploading of numerical as well as gel/blot image source data; for details see: embopress.org/page/journal/14602075/authorguide#sourcedata

At EMBO Press, we ask authors to provide source data for the main manuscript figures. Our source data coordinator will contact you to discuss which figure panels we would need source data for and will also provide you with helpful tips on how to upload and organize the files.

In the interest of ensuring the conceptual advance provided by the work, we recommend submitting a revision within 3 months (29th Jul 2024). Please discuss the revision progress ahead of this time with the editor if you require more time to complete the revisions. Use the link below to submit your revision:

Link Not Available

Referee #1:

The authors have addressed my minor comments. I recommend publication.

Referee #2:

This manuscript is substantially improved and, along with the accompanying manuscript, will be an interesting addition to the field.

The one remaining point needing clarification has to do with the quantification and statistics on Fig. 4C. The improved images and quantification strengthen their claims substantially. However, the statistical comparison that is being made is confusing based on the figure. Is each condition shown (siPARP9, siDTX3L, and siPARP14) statistically different than the siCTRL? Or is the group of three experimental conditions that is statistically different than siCTRL, which is what I infer based on the way it is currently presented with a single line and a single p-value? It would be better to show the exact pairwise comparisons (including

their relevant p-value groupings by asterisk) rather than the grouped bar. It would also be helpful to either expand the y-axis or directly report the average of each condition, since it is currently very difficult to assess how the red average line differs between siCTRL and siDTX3L and siPARP14. Even if the siDTX3L and/or siPARP14 conditions are not as striking (or are not statistically significant) as the siPARP9 condition, being transparent with the data is important and does not invalidate their overall conclusions.

Response to reviewer 2

This manuscript is substantially improved and, along with the accompanying manuscript, will be an interesting addition to the field.

We thank the reviewer for these comments.

The one remaining point needing clarification has to do with the quantification and statistics on Fig. 4C. The improved images and quantification strengthen their claims substantially. However, the statistical comparison that is being made is confusing based on the figure. Is each condition shown (siPARP9, siDTX3L, and siPARP14) statistically different than the siCTRL? Or is the group of three experimental conditions that is statistically different than siCTRL, which is what I infer based on the way it is currently presented with a single line and a single p-value? It would be better to show the exact pairwise comparisons (including their relevant p-value groupings by asterisk) rather than the grouped bar. It would also be helpful to either expand the y-axis or directly report the average of each condition, since it is currently very difficult to assess how the red average line differs between siCTRL and siDTX3L and siPARP14. Even if the siDTX3L and/or siPARP14 conditions are not as striking (or are not statistically significant) as the siPARP9 condition, being transparent with the data is important and does not invalidate their overall conclusions.

We apologise for the less-than-optimal visualisation of our data. We have taken into account the comments and have changed the figure in several ways. Firstly, we have increased the overall height of the graph allowing for better spread of the datapoints for better visualisation of the differences between conditions. Secondly, we now improve the visualisation of the statistical comparison making it clear that each knockdown is significantly different to the control. Finally, in the figure legend we have directly reported the mean for each condition. We attach below the improved figure and legend.

(C) Quantification of number of ubiquitin foci that also show staining for ADPr from (B). Statistical analysis was determined using a Kruskal Wallis test with Bonferroni correction followed by post-hoc Dunn's test. Asterisks indicate statistical significance compared to control (**: $p < 0.01$; ****: $p < 0.0001$).

Data information: Graph in C shows number of foci per individual cell 279 cells per condition. The red line indicates the mean (siCTRL: 0.266; siPARP9: 1.688; siDTX3L: 0.182; siPARP14: 0.019). Data are representative of a minimum of three (A) or five (B-C) independent replicates. Source data are available online for this figure.

Dr. Ivan Ahel
University of Oxford
Sir William Dunn School of Pathology
South Parks Road
Oxford OX1 3RE
United Kingdom

8th May 2024

Re: EMBOJ-2023-115820R1
PARP14 and PARP9/DTX3L regulate interferon-induced ADP-ribosylation

Dear Ivan,

Thank you for submitting your final revised manuscript for our consideration. I am pleased to inform you that we have now accepted it for publication in The EMBO Journal.

Yours sincerely,

Hartmut
